systems biology/mathematical modelling/health and disease and epidemiology

dengue, optimal control, sterile insect technique, migration, network

**Author for correspondence:**
Thomas Rawson
e-mail: thomas.rawson@zoo.ox.ac.uk

# Optimal control approaches for combining medicines and mosquito control in tackling dengue

Thomas Rawson[1], Kym E. Wilkins[2] and Michael B. Bonsall[1]

[1]Mathematical Ecology Research Group, Department of Zoology, University of Oxford, Oxford OX1 3PS, UK
[2]School of Mathematical Sciences, The University of Adelaide, Adelaide, South Australia 5005, Australia

TR, 0000-0001-8182-4279; MBB, 0000-0003-0250-0423

Dengue is a debilitating and devastating viral infection spread by mosquito vectors, and over half the world's population currently live at risk of dengue (and other flavivirus) infections. Here, we use an integrated epidemiological and vector ecology framework to predict optimal approaches for tackling dengue. Our aim is to investigate how vector control and/or vaccination strategies can be best combined and implemented for dengue disease control on small networks, and whether these optimal strategies differ under different circumstances. We show that a combination of vaccination programmes and the release of genetically modified self-limiting mosquitoes (comparable to sterile insect approaches) is always considered the most beneficial strategy for reducing the number of infected individuals, owing to both methods having differing impacts on the underlying disease dynamics. Additionally, depending on the impact of human movement on the disease dynamics, the optimal way to combat the spread of dengue is to focus prevention efforts on large population centres. Using mathematical frameworks, such as optimal control, are essential in developing predictive management and mitigation strategies for dengue disease control.

## 1. Introduction

Approximately 3.9 billion people live at risk of contracting dengue [1], a viral infection transmitted by *Aedes* mosquitoes [2]. Vectors become infected upon ingesting blood from an infected host, and then pass on the disease by biting susceptible individuals. Current

estimates suggest that 390 million people are infected with dengue each year, of which 96 million display clinical symptoms [3]. Owing to the method of transmission, the disease is prevalent in tropical and subtropical climates across the globe, most commonly in urban environments, where dense populations and stagnant water sources create ideal mosquito breeding grounds [4]. The steady increase in sudden outbreaks and an increase in the epidemic potential highlights dengue as a serious, and growing, concern for public health [5].

Dengue has atypical epidemiology. Five serotypes have so far been identified, all of which are capable of inducing the same critical health problems [6] (note that the fifth serotype was only recently discovered, and much of the existing literature still refers to only four serotypes). When a susceptible individual contracts dengue, they enter an infectious state followed by a brief period of cross-immunity to all serotypes once they have recovered. However, this immune period wanes and individuals become highly susceptible to the other dengue serotypes, with an astonishingly high increase of morbidity and mortality. This effect is known as antibody-dependent enhancement [7].

Although vaccine development continues to advance [8,9], currently, the primary method of suppressing dengue is through vector control. By limiting mosquito densities, transmission of the virus is directly reduced as a result of the population sizes of vectors falling below an entomological threshold. While vector control has focussed on insecticides, an increasingly promising approach is the use of genetically modified mosquitoes. Building on approaches developed for agricultural pest management, the sterile insect technique (SIT), whereby modified, infertile male insects are released into the environment to mate with wild-types, has been proven to reduce the number of live young produced, and subsequently suppress total population size. This SIT method has been a tremendous success in some situations [10], but many studies have begun to show the limitations of the technique owing to distribution and dispersal issues [11]. Genetic-based methods similar to the SIT method have been developed whereby lethal genes switch on at key stages in the insect's life cycle to induce mortality [12]. The overall effect is to then reduce the mosquito population size, albeit in a self-limiting way, as once mass releases stop, the control effect diminishes and populations may resurge and disease control may be lost.

As a result, combining these vector intervention methods to combat dengue outbreaks continues to be explored. One particularly exciting combination is with the increasing availability of dengue vaccinations being produced to aid disease control efforts.

In early 2016, the first dengue vaccine, Dengvaxia, was registered for use in several countries across southeast Asia [13]. Ongoing clinical results, however, show that the vaccine currently displays limited efficacy [14], suggesting that it may not be feasible to tackle the crisis of this disease through vaccination alone. Indeed Robinson & Durbin [15] report that while vaccines currently may reduce the immediate dengue burden, population-level impact will probably be limited over time. While a recent published model of dengue dynamics [16] investigates how combined control interventions for dengue on small networks could be developed, optimal strategies for adequately controlling dengue epidemics remain difficult to determine. Here, we address this by developing on the framework [16] as a basis for investigating the effectiveness of each approach (vaccination, vector control), seeking an optimal strategy to managing dengue outbreaks under varying circumstances.

In the next section, we introduce the mathematical model and the optimal control formulation. We then investigate how the dynamical impact of these disease prevention methods affects the spread of disease, and how they may be best combined and employed across differing scenarios. In the discussion, we highlight the potential of the work for explaining the interplay between these two imperfect methods of disease prevention.

# 2. Methods

## 2.1. Governing ordinary differential equations

For the dengue disease management, our optimal control problem has been developed around a mathematical framework presented by Hendron & Bonsall [16]. The model describes the population dynamics of both vectors and hosts by describing the complex epidemiology associated with dengue infections. These states are summarized in table 1, and schematics of this epidemiology are illustrated in figure 1. For the dengue epidemiology, the model framework only considers two potential serotypes (labelled $A$ and $B$). As well as simplifying the necessary mathematics, it is unusual for more than two serotypes to be in circulation in a small population at one time [17]. Dengue infections lead to complex

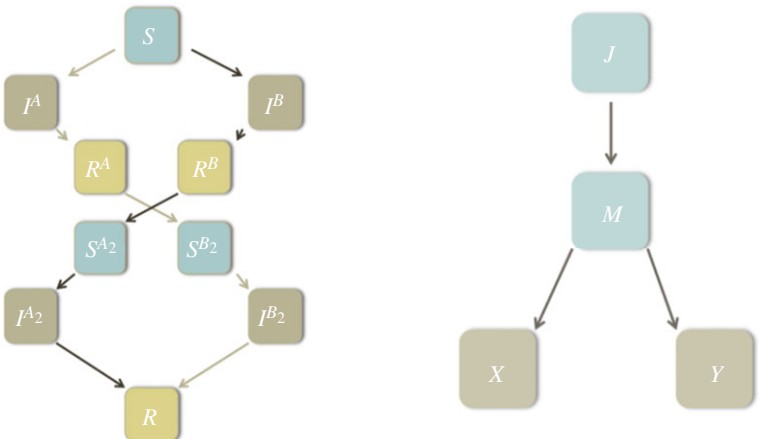

**Figure 1.** Epidemiological frameworks. (*a*) Schematic illustration of the epidemiological framework for two serotypes of dengue circulating in a host population. (*b*) Schematic illustration of the epidemiological framework for the mosquito vector dynamics. All state variables are defined in table 1.

**Table 1.** Definitions of the state variables considered in the model framework following Hendron & Bonsall [16].

| state variable | definition |
| --- | --- |
| $S$ | hosts susceptible to either serotype |
| $I^A$ | hosts infected with serotype $A$, as a primary infection |
| $R^A$ | hosts recovered from serotype $A$ |
| $S^{B_2}$ | hosts now susceptible to serotype $B$, as a secondary infection |
| $I^{B_2}$ | hosts infected with serotype $B$, as a secondary infection |
| $I^B$ | hosts infected with serotype $B$, as a primary infection |
| $R^B$ | hosts recovered from serotype $B$ |
| $S^{A_2}$ | hosts now susceptible to serotype $A$, as a secondary infection |
| $I^{A_2}$ | hosts infected with serotype $A$, as a secondary infection |
| $R$ | hosts recovered from all serotypes |
| $J$ | juvenile vectors |
| $M$ | mature vectors |
| $X$ | vectors infected with serotype $A$ |
| $Y$ | vectors infected with serotype $B$ |

individual host patterns of infection, and two distinct serotypes can create two possible routes of infection (as shown in figure 1).

An initial susceptible pool of hosts ($S$) transfers to a state of primary infection ($I^A$ or $I^B$), as individuals are bitten by a mosquito infected with the respective serotype. Eventually, they recover from this primary infection, entering a brief period of cross-immunity ($R^A$ or $R^B$). While remaining immune to the primary serotype they were infected with, hosts then enter a state of susceptibility to a secondary infection ($S^{A_2}$ or $S^{B_2}$). They may then be bitten and infected with the other serotype ($I^{A_2}$ or $I^{B_2}$), before eventually recovering again into a period of complete recovery ($R$).

Mosquitoes display a simpler dynamic (figure 1*b*). Upon hatching, mosquitoes start in a juvenile state ($J$), where they cannot yet bite a host. Eventually, they mature into an adult state ($M$) susceptible to dengue virus infection. Upon biting an infected host, these mature mosquitoes then enter a state of infection with either serotype $A$ or $B$ ($X$ or $Y$, respectively). It has been shown experimentally that *Aedes* mosquitoes will resist infection of a second serotype [18], hence our assumption that mosquitoes may be infected with serotype $A$ or $B$, but not both.

**Table 2.** Model constants and control variables.

| expression | description | value |
|---|---|---|
| $b_1$ | conversion rate of $S$ to $I^{A/B}$ | 0.38 [19] |
| $b_2$ | conversion rate of $S^{A_2/B_2}$ to $I^{A_2/B_2}$ | 0.57 [19] |
| $\alpha_1$ | mortality rate of hosts with a primary infection | $4.57 \times 10^{-4}$ [20] |
| $\alpha_2$ | mortality rate of hosts with a secondary infection | $8.33 \times 10^{-3}$ [21] |
| $\rho_1$ | recovery rate from a primary infection | 0.17 [19] |
| $\rho_2$ | recovery rate from a secondary infection | 0.17 [19] |
| $\eta$ | rate of loss of cross-immunity | $8.33 \times 10^{-3}$ [19] |
| $a$ | biting rate of adult mosquitoes | 0.5 [19] |
| $g$ | eggs laid by adult mosquitoes per day | 0.7 [19] |
| $\delta$ | density-dependent parameter | 0.1 [22] |
| $\phi$ | juvenile mosquito maturation rate | 0.0541 [19] |
| $\mu$ | adult mosquito mortality rate | 0.0741 [19] |
| $c$ | conversion rate into dengue vectors | 0.38 [19] |
| $\xi$ | (im)perfect vaccine switch | 0 or 1 |
| $\psi_i$ | maximum possible number of released modified mosquitoes | $10 \times Z_0$ |
| $\varepsilon$ | gene lethality | 0.9 |
| $m_S$ | proportion of susceptible population that migrates | 0.1 |
| $m_I$ | proportion of primary infected population that migrates | 0.1 |
| $m_{I_2}$ | proportion of secondary infected population that migrates | 0.01 |
| $m_R$ | proportion of recovered population that migrates | 0.1 |
| $u_1(t)$ | proportion of susceptible population being vaccinated | $0 \leq u_1 \leq 1$ |
| $u_2(t)$ | proportion of total possible SIT modified mosquitoes released | $0 \leq u_2 \leq 1$ |

A key feature of the model framework is its description of human movement on a small network [16]. Nodes (settlements) are considered with their own distinct host/vector populations and disease dynamics, but with the assumption that hosts may travel from their home settlement to another location (perhaps a place of work). Primarily, we will focus on a 'hub-spoke' network, where several satellite town settlements have a percentage of their populations migrate into a central city settlement temporarily, before returning to their home settlement.

The governing equations follow Hendron & Bonsall [16] (where a more detailed description of the model is available). The dynamics of susceptible hosts in settlement $i$ (where $i \in \{1, 2, \ldots, n\}$, $n$ being the total number of settlements) can be described by

$$\frac{dS_i}{dt} = -\left(1 - \sum_{\substack{j=1 \\ j \neq i}}^{n} w_{ji} m_S\right) \frac{S_i a b_1 (X_i + Y_i)}{N_i + \Omega_i} - \sum_{\substack{j=1 \\ j \neq i}}^{n} w_{ji} m_S \frac{S_i a b_1 (X_j + Y_j)}{N_j + \Omega_j} - u_1^i S_i, \tag{2.1}$$

where the first term on the right hand side (RHS) of equation (2.1) represents the members of $S_i$ who are removed from this state by being bitten by an infected mosquito while in their home settlement. The second term represents those hosts who are bitten by an infected mosquito while in a different settlement they have migrated to. Finally, the third term represents those who have left the class by being vaccinated, where the variable $u_1^i$ represents the proportion of the population of $S_i$ who are being vaccinated. As such, $0 \leq u_1^i \leq 1$. $w_{ji}$ is a switch variable where either $w_{ji} = 1$, meaning there exists migration from settlement $i$ to settlement $j$, or $w_{ji} = 0$, meaning there is no migration from $i$ to $j$. $m_S$ is a constant, defining the proportion of a settlement's susceptible population that will migrate. All rate constants are defined in table 2. Note that the values used, as shown in table 2, are taken from experimental works with the intention of using values of accurate magnitude in relation to one

another. The timescale of the model is presented where $t$ is the number of days from initialization. $N_i$ is the total host population of settlement $i$, $Z_i$ is the total adult mosquito population of settlement $i$, and $\Omega_i$ is the total net movements of hosts into, and out of, settlement $i$. Formally,

$$N_i = S_i + I_i^A + R_i^A + S_i^{B_2} + I_i^{B_2} + I_i^B + R_i^B + S_i^{A_2} + I_i^{A_2} + R_i,$$
$$Z_i = M_i + X_i + Y_i$$

and
$$\Omega_i = \sum_{\substack{j=1 \\ j \neq i}}^{n} w_{ij}(m_S S_j + m_I(I_j^A + I_j^B) + m_R(R_j^A + R_j^B + S_j^{A_2} + S_j^{B_2} + R_j) + m_{I_2}(I_j^{A_2} + I_j^{B_2}))$$
$$- w_{ji}(m_S S_i + m_I(I_i^A + I_i^B) + m_R(R_i^A + R_i^B + S_i^{A_2} + S_i^{B_2} + R_i) + m_{I_2}(I_i^{A_2} + I_i^{B_2})).$$

As such, $(N_i + \Omega_i)$ represents the total population of site $i$ during the day, as $N_i$ represents the native human population, and $\Omega_i$ deducts those who have migrated out of settlement $i$ during the day, and includes those who have migrated in for the day from other settlements. This allows for the simulation of the event where an individual host could migrate to a settlement, contract dengue, return home, and then infect their native vector populations. Note that infection rates above were divided by $(N_i + \Omega_i)$ in the differential equation for $S_i$ to factor in the likelihood of a vector infecting a susceptible individual, and not another member of the population belonging to the $(N_i + \Omega_i)$ class. Logically, a smaller susceptible population implies a lower rate of infection.

The governing dynamics of individuals infected with serotype $A$, $I^A$, are described by

$$\frac{dI_i^A}{dt} = \left(1 - \sum_{\substack{j=1 \\ j \neq i}}^{n} w_{ji} m_S\right) \frac{S_i a b_1 X_i}{N_i + \Omega_i} + \sum_{\substack{j=1 \\ j \neq i}}^{n} w_{ji} m_S \frac{S_i a b_1 X_j}{N_j + \Omega_j} - I_i^A(\alpha_1 + \rho_1). \tag{2.2}$$

Similar to equation (2.1), the first term on the RHS represents those individuals who became infected while in their home settlement, and the second term represents those who become infected in another settlement. The third term represents those who leave the class either by recovering, or by death.

Equation (2.3) shows the behaviour of individuals recovering from serotype $A$, $R_i^A$:

$$\frac{dR_i^A}{dt} = \rho_1 I_i^A - \eta R_i^A. \tag{2.3}$$

People enter the class by recovering from the infection (at rate $\rho_1$), and leave after the period of cross-immunity fades (at rate $\eta$). Equations (2.4) and (2.5) explain the secondary infections, and follow the same logic as equations (2.1) and (2.2):

$$\frac{dS_i^{B_2}}{dt} = \eta R_i^A - \left(1 - \sum_{\substack{j=1 \\ j \neq i}}^{n} w_{ji} m_R\right) \frac{S_i^{B_2} a b_2 Y_i}{N_i + \Omega_i} - \sum_{\substack{j=1 \\ j \neq i}}^{n} w_{ji} m_R \frac{S_i^{B_2} a b_2 Y_j}{N_j + \Omega_j} \tag{2.4}$$

and

$$\frac{dI_i^{B_2}}{dt} = \left(1 - \sum_{\substack{j=1 \\ j \neq i}}^{n} w_{ji} m_R\right) \frac{S_i^{B_2} a b_2 Y_i}{N_i + \Omega_i} + \sum_{\substack{j=1 \\ j \neq i}}^{n} w_{ji} m_R \frac{S_i^{B_2} a b_2 Y_j}{N_j + \Omega_j} - I_i^{B_2}(\alpha_2 + \rho_2). \tag{2.5}$$

Equations (2.6)–(2.10) describe the alternate route of infection by initially acquiring serotype $B$ (figure 1a). The only difference is in equation (2.7), where the final term on the RHS allows for the possibility that vaccination only protects against one particular serotype, hence moving an individual from $S$ to $R^B$

instead of $R$:

$$\frac{\mathrm{d}I_i^B}{\mathrm{d}t} = \left(1 - \sum_{\substack{j=1 \\ j \neq i}}^{n} w_{ji}m_S\right)\frac{S_i ab_1 Y_i}{N_i + \Omega_i} + \sum_{\substack{j=1 \\ j \neq i}}^{n} w_{ji}m_S \frac{S_i ab_1 Y_j}{N_j + \Omega_j} - I_i^B(\alpha_1 + \rho_1),$$ (2.6)

$$\frac{\mathrm{d}R_i^B}{\mathrm{d}t} = \rho_1 I_i^B - \eta R_i^B + (1 - \xi)u_1^i S_i,$$ (2.7)

$$\frac{\mathrm{d}S_i^{A_2}}{\mathrm{d}t} = \eta R_i^B - \left(1 - \sum_{\substack{j=1 \\ j \neq i}}^{n} w_{ji}m_R\right)\frac{S_i^{A_2} ab_2 X_i}{N_i + \Omega_i} - \sum_{\substack{j=1 \\ j \neq i}}^{n} w_{ji}m_R \frac{S_i^{A_2} ab_2 X_j}{N_j + \Omega_j}$$ (2.8)

and

$$\frac{\mathrm{d}I_i^{A_2}}{\mathrm{d}t} = \left(1 - \sum_{\substack{j=1 \\ j \neq i}}^{n} w_{ji}m_R\right)\frac{S_i^{A_2} ab_2 X_i}{N_i + \Omega_i} + \sum_{\substack{j=1 \\ j \neq i}}^{n} w_{ji}m_R \frac{S_i^{A_2} ab_2 X_j}{N_j + \Omega_j} - I_i^{A_2}(\alpha_2 + \rho_2).$$ (2.9)

Finally, the ordinary differential equation (ODE) for the final, fully recovered, class is

$$\frac{\mathrm{d}R_i}{\mathrm{d}t} = \rho_2(I_i^{A_2} + I_i^{B_2}) + \xi u_1^i S_i.$$ (2.10)

Note again the possibility that vaccination can move susceptible hosts directly to this class. Choosing $\xi$ to be 0 or 1 dictates whether the vaccination moves susceptible hosts to a fully protected class, $R$ or only protected against one serotype ($R^B$).

The final four ODEs describe the dynamic behaviour of the vector populations, as shown in figure 1$b$. The mosquito dynamics are divided into juvenile and adult states. The juvenile dynamics are described by

$$\frac{\mathrm{d}J_i}{\mathrm{d}t} = gZ_i\left(\frac{Z_i}{Z_i + \psi_i u_2^i} + (1 - \epsilon)\frac{\psi_i u_2^i}{Z_i + \psi_i u_2^i}\right) - J_i \ln(1 + \delta J_i) - \phi J_i,$$ (2.11)

where the juvenile mosquitoes are born at a rate $g$, while being slowed by the amount of modified mosquitoes ($Z_i/(Z_i + \psi_i u_2^i) + (1 - \epsilon)\psi_i u_2^i/(Z_i + \psi_i u_2^i)$) in the population that compete with mating wild-types. $u_2^i$ is a variable that describes the proportion of the total possible matings by SIT mosquitoes, $\psi_i$, that have been released in settlement $i$. $\epsilon$ represents the actual effectiveness of the SIT, dictating how many of the modified mosquitoes have been correctly made infertile. $\epsilon = 1$ represents a fully efficacious mosquito sterilization treatment, while $\epsilon = 0$ would signify that none of the released mosquitoes were successfully sterilized. Note therefore that, if no SIT mosquitoes are released, and $u_2^i = 0$, then ($Z_i/(Z_i + \psi_i u_2^i) + (1 - \epsilon)(\psi_i u_2^i/(Z_i + \psi_i u_2^i))$) reduces to ($Z_i/Z_i + (1 - \epsilon)(0/Z_i)) = 1$. Hence, with no SIT mosquito release, the birth rate of juvenile mosquitoes ($J_i$) is simply $gZ_i$. Similarly, if it was set that $\epsilon = 0$, meaning that the modified mosquito treatment was entirely unsuccessful, then ($Z_i/(Z_i + \psi_i u_2^i) + (1 - \epsilon)(\psi_i u_2^i/(Z_i + \psi_i u_2^i))$) reduces to (($Z_i + \psi_i u_2^i)/(Z_i + \psi_i u_2^i)) = 1$, which similarly results in a growth rate of solely $gZ_i$. The second term on the RHS, ($J_i\ln(1 + \delta J_i)$), reflects juvenile mosquito mortality owing to density-dependent intraspecific competition. This form of monotonic growth to a fixed point is well discussed [22]. The final term ($\phi J_i$) is the juvenile maturation rate.

Equation (2.12) describes the adult mosquito population:

$$\frac{\mathrm{d}M_i}{\mathrm{d}t} = -\frac{acM_i}{N_i + \Omega_i}\left((I_i^A + I_i^B)\left(1 - \sum_{\substack{j=1 \\ j \neq i}}^{n} w_{ji}m_I\right) + (I_i^{A_2} + I_i^{B_2})\left(1 - \sum_{\substack{j=1 \\ j \neq i}}^{n} w_{ji}m_{I_2}\right)\right)$$
$$- \frac{acM_i}{N_i + \Omega_i}\left(\sum_{\substack{j=1 \\ j \neq i}}^{n} w_{ij}(m_I(I_j^A + I_j^B) + m_{I_2}(I_j^{A_2} + I_j^{B_2}))\right) + \phi J_i - \mu M_i,$$ (2.12)

where the first term on the RHS represents a mosquito becoming infected upon biting an infected host native to settlement $i$, while the second term represents biting an infected host who has migrated from a different settlement. The third term is the rate of maturation of the juveniles entering the class $M_i$, and finally again the natural death rate of the uninfected vectors.

Lastly, equations (2.13) and (2.14) describe the dynamics for mosquitoes infected with serotypes $A$ and $B$, respectively. Their form is comparable to equation (2.12), but only the infected hosts with the associated serotype are considered:

$$\frac{\mathrm{d}X_i}{\mathrm{d}t} = \frac{acM_i}{N_i + \Omega_i}\left(I_i^A(1 - \sum_{\substack{j=1 \\ j \neq i}}^{n} w_{ji}m_I) + I_i^{A_2}(1 - \sum_{\substack{j=1 \\ j \neq i}}^{n} w_{ji}m_{I_2})\right)$$
$$+ \frac{acM_i}{N_i + \Omega_i}\sum_{\substack{j=1 \\ j \neq i}}^{n} w_{ij}(m_I I_j^A + m_{I_2} I_j^{A_2}) - \mu X_i$$

(2.13)

and

$$\frac{\mathrm{d}Y_i}{\mathrm{d}t} = \frac{acM_i}{N_i + \Omega_i}\left(I_i^B(1 - \sum_{\substack{j=1 \\ j \neq i}}^{n} w_{ji}m_I) + I_i^{B_2}(1 - \sum_{\substack{j=1 \\ j \neq i}}^{n} w_{ji}m_{I_2})\right)$$
$$+ \frac{acM_i}{N_i + \Omega_i}\sum_{\substack{j=1 \\ j \neq i}}^{n} w_{ij}(m_I I_j^B + m_{I_2} I_j^{B_2}) - \mu Y_i.$$

(2.14)

Because the model is measured over a timescale of days, new host birth events are negligible, and we consider a fixed starting human population. This also means that our final cost function will not be affected by the simulated time period, as we consider the total time until all hosts have reached a model state whereby they can no longer be aided by control measures.

## 2.2. Optimal control problem

Disease dynamics are readily amenable to optimal control problems; often the key epidemiological question is—what is the optimal choice of disease intervention measures (vaccines, vector-control methods) to minimize the number of infected individuals while also trying to minimize the economic cost of such measures. Traditional SIR networks are well formulated for optimal control approaches, and are commonly used in identifying optimal vaccination programmes [23], often in parallel with other methods and/or treatments [24]. One simple example is presented by Yusuf & Benyah [25].

A more in-depth introduction and explanation of optimal control problems is given in the electronic supplementary material, S1 appendix. In short, these are problems whereby a certain objective is desired, and is limited by certain constraints. In our particular case, we wish to minimize the number of infected individuals; however, we are limited in some form by how much we can apply disease-prevention methods, both logistically and economically. An optimal control problem is most commonly formulated around a model of differential equations, explaining the interaction between several variables of interest. These variables are denoted as either **state variables**, variables which cannot be directly impacted, or as **control variables**, variables that can be directly altered. In our case, our control variables are $u_1^i$, the proportion of individuals in settlement $i$ being vaccinated, and $u_2^i$ the proportion of the total possible SIT mosquitoes, $\psi_i$, being released into settlement $i$. Owing to the ongoing research and development of the Dengvaxia vaccine, we do not attempt to simulate the specific attributes of this particular vaccine. We instead consider application of the vaccine to the naive population, representing an idealized aim of this sort of treatment. All other variables are state variables, as we cannot directly change these variables, and they instead change dynamically as we alter our control variables.

For this dengue control problem, we are considering $14n$ state variables and $2n$ control variables, where $n$ is the total number of settlements being considered. A desired objective is then to either maximize or minimize a certain number of state variables, while also trying to minimize the total magnitude of the control variables themselves. This desired outcome is compiled as an objective functional; a total expression that we wish to minimize. Combined with the fact that we clearly wish to minimize the number of infected individuals, our objective functional to be considered is,

$$\min \int_0^{t_f} \sum_{i=1}^{n} C_1(I_i^{A^2} + I_i^{B^2} + I_i^{A_2^2} + I_i^{B_2^2}) + C_2 u_1^{i^2} + C_3 u_2^{i^2},$$

(2.15)

where $C_1$, $C_2$ and $C_3$ are the daily costs associated with treating those infected with dengue, vaccinating a population and releasing SIT mosquitoes, respectively. Estimating realistic values for these cost parameters is inherently difficult, and hence the model is run for varying cost values to investigate the impact that choice of cost has on the results. Note also that we seek to minimize the number of infected individuals ($I_i^{A2}$, $I_i^{B2}$, $I_{i_2}^{A2}$, $I_{i_2}^{B2}$) as opposed to disease-induced deaths. Our model formulation shows that a set proportion ($\alpha_1$ and $\alpha_2$) of the infected classes will die as a result of the disease, representing a linear relationship between the infected population and the disease-induced mortality. As such, seeking to minimize the infected population provides the same result as seeking to minimize host mortality. We seek the optimal choices of $u_1^i$ and $u_2^i$ so as to minimize the cost function (equation (2.15)). The decision to square the variables in our cost function (equation (2.15)) represents a nonlinear cost. While this simplifies the resulting mathematical analysis, more importantly, it also increases the range of information our optimal control system can provide. When using a linear control constraint, the optimality condition does not produce a characterization of the control, and as such can only return a piecewise continuous optimal control variable, known as a 'bang-bang control'. More information can be found in Lenhart & Workman [26]. In short, using a quadratic control in equation (2.15) provides us with a more versatile cost framework for analysis. Elsewhere, the results of choice of cost function on optimal control outcomes for vector-borne disease control has been investigated [27].

The choice of the objective functional is a key assumption in an optimal control problem, as we impose structure upon the system. As such, it is important to mention the impact that different choices of formulation can have on the final results. One simplification we make is to combine both primary and secondary infections under the same cost value, $C_1$. This is despite secondary infections resulting in a more severe host response as mentioned above. This decision was made owing to, firstly, a lack of information as to the likely increase in hospitalization costs within an already wide spectrum of treatment costs between cases. Secondly, the model dynamics show that secondary infections will always be less populous than primary infections, and the impact of control measures within our case studies reduce secondary infections to a near-negligible level. Thirdly, the underlying dynamics of the objective functional represents a balance between minimizing infected individuals and minimizing the application of control measures, creating a 'tug-of-war' effect. As such, and as model simulations showed, the dynamical behaviour of our optimal solution reacted such that increasing $C_1$ would be dynamically equivalent to decreasing $C_2$ or $C_3$. All of our case studies included feature variations on cost parameters to display this feature.

As with all optimal control problems, we begin by formulating the Hamiltonian, $H = F(t, x, u) + \lambda(t)f(t, x, u)$, where $F(t, x, u)$ is our cost function (equation (2.15)), $\lambda(t)$ our adjoint functions and $f(t, x, u)$ our governing ODEs (equations (2.1)–(2.14) above). We give our specific Hamiltonian in full in the electronic supplementary material, S2 appendix. Adjoint equations are defined as $d\lambda/dt = -\partial H(t, x^*(t), u^*(t), \lambda(t))/\partial x$. The adjoint equations represent the costs that would be incurred by breaking the constraints of the minimization problem. For notational ease, our adjoint functions will be written as $\lambda_j^i$, where $i$ refers to the settlement of the associated state variable, and $j$ to which of the 14 state variables it is associated with. For example, $\lambda_2^i$ is the adjoint function associated with $I_i^A$. As such we will have $14n$ adjoint terms in total, $n$ being the number of settlements. By using the optimality condition, $\partial H/\partial u = 0$, at the optimal state, we can acquire an expression for our control variables in terms of our adjoints and state variables. Namely,

$$u_1^i = \frac{S_i}{2C_2}(\lambda_1^i - \lambda_7^i(1 - \epsilon) - \lambda_{10}^i \xi)$$

and

$$u_2^i = \frac{1}{3}\left( \frac{-2Z_i}{\psi_i} + \left( \frac{Z_i^3}{\psi_i^3} + \frac{27\lambda_{11}^i \epsilon g Z_i^2}{4C_3 \psi_i} + \sqrt{\frac{27\lambda_{11}^i \epsilon g Z_i^5}{2C_3 \psi_i^4} + \frac{729(\lambda_{11}^i \epsilon g)^2 Z_i^4}{16C_3^2 \psi_i^2}} \right)^{1/3} \right)$$

$$+ \left( \frac{Z_i^2}{3\psi_i^2}\left( \frac{Z_i^3}{\psi_i^3} + \frac{27\lambda_{11}^i \epsilon g Z_i^2}{4C_3 \psi_i} + \sqrt{\frac{27\lambda_{11}^i \epsilon g Z_i^5}{2C_3 \psi_i^4} + \frac{729(\lambda_{11}^i \epsilon g)^2 Z_i^4}{16C_3^2 \psi_i^2}} \right)^{-1/3} \right).$$

Finally, all that remains is to provide the adjoint equations. Owing to the large number of variables all interacting within the Hamiltonian, a key assumption was made to simplify the derivation of the adjoint equations. We always assume a *spoke-and-hub* network, whereby a central settlement (*the hub*), which we shall always define as the settlement $i = 1$, has hosts migrate into it from the other, satellite, settlements (*the spokes*). No hosts will migrate out from the central city settlement, and no hosts will

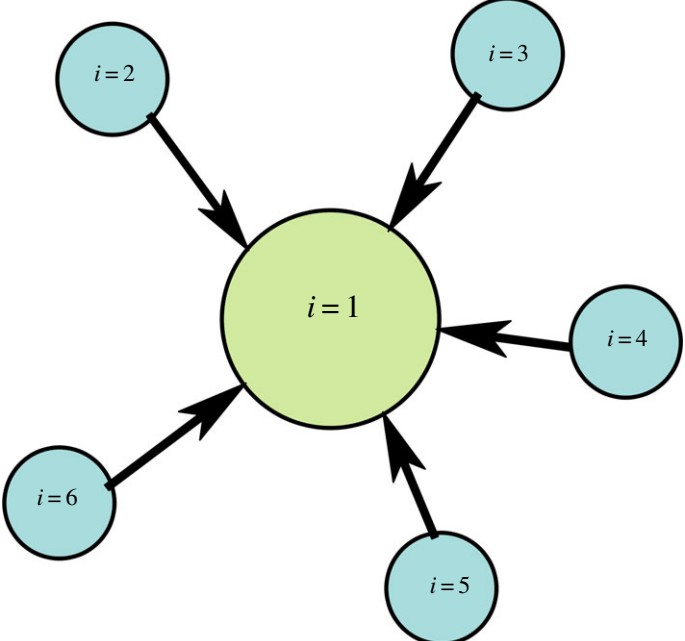

**Figure 2.** Network example. A spoke-and-hub network example ($n = 6$ settlements) used to investigate dengue disease dynamics. Movement only occurs from the outer settlements (spoke) into the central settlement (hub).

migrate from the smaller, outside settlements to one another. Note however that this does not represent a permanent migration, instead just a temporary change in location. All hosts have a designated 'home node' which they always return to, but may migrate to another node, creating another opportunity to become infected in a different location. This is shown in figure 2 and the adjoint equations are given in full in the electronic supplementary material, S3 appendix.

## 3. Results

We present results from a series of case studies. For the first three cases, we consider a situation with only a single settlement. Unless stated otherwise, the values of the constants remain at that defined in table 2. Unless stated otherwise, for the numerics, we use the following initial conditions: $S_0 = 2\,000\,000$, $I_0^A = 1$, $I_0^B = 1$, $J_0 = 75\,000\,000$, $Z_0 = 4\,000\,000$, $M_0 = 3\,996\,000$, $X_0 = 2000$ and $Y_0 = 2000$. State variables not specified are initialized at 0. All computations were undertaken in MATLAB and code is available here: https://osf.io/vspxe/ [28].

### 3.1. Case 1: vector-control (sterile insect technique) only

We start by considering a case without vaccines within just one settlement (i.e. we define $u_1$ to always be equal to 0 for this case). Carrasco *et al.* [29] used data from the Singapore Ministry of Health to estimate the hospitalization costs per case per day. This was found to be modelled by a truncated normal distribution with a mean of \$431 (US dollars; USD) and standard deviation \$597 (USD), bounded below at 0. This large variance in cost is owing to the differences in symptomatic expression among hosts. With this in mind, we choose $C_1 = \$510$ (USD), the zero-bounded mean of the above distribution, as an estimate of expected healthcare costs. This distribution was calculated as an average for each hospitalized case, and while not all cases of dengue may result in hospitalization, this captures our inherent desire to minimize human illness, by interpreting each host infection as a potential for hospitalization. For $C_3$, Alphey *et al.* [19] sets the total costs of setting up and maintaining a 5-year SIT programme as ranging from 0.342 to 1.109 million USD. Figure 3 below shows a sensitivity analysis in choosing values of $C_3$ ranging from $\frac{342\,000}{5(365)} \approx 190$ to $\frac{1\,109\,000}{5(365)} \approx 610$. We do not perform sensitivity analyses upon initial model parameters, as this was undertaken in the original

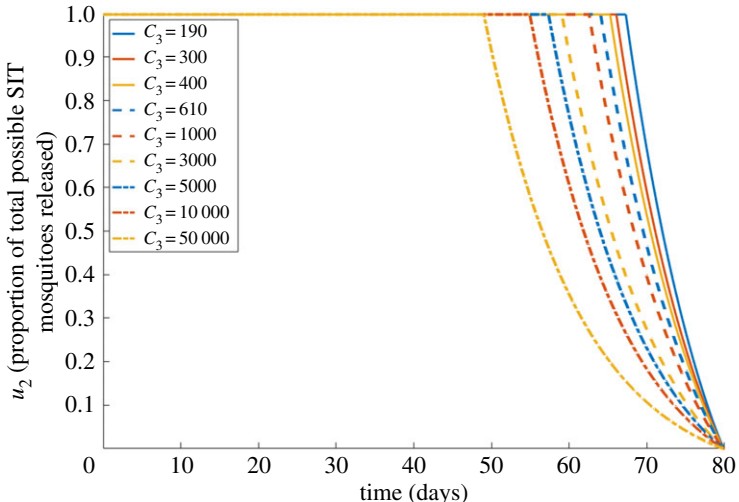

**Figure 3.** Case 1 results. Optimal choices through time for vector control ($u_2$) for different values of SIT programme costs ($C_3$). Other costs are held constant: $C_1 = 510$. Increasing costs lead to earlier reductions in the proportion of modified mosquitoes released to control infections.

publication [16], revealing no impact to model dynamics. Upon seeing how closely these results clustered, even larger values of $C_3$ were also considered.

Figure 3 shows that, for all control costs ($C_3$), it is always optimal to apply the maximum possible SIT-mosquito release at the start of the outbreak. The control is then later reduced, at earlier points in time for larger values of $C_3$. This reflects how, for higher control costs, it becomes more economically optimal to start gradually reducing control measures. Note that these optimal decreases to the control variable always occur at the end of the disease outbreak, as by this point vector populations will have already been reduced by the earlier control efforts. As such, there is a reduced risk of infection at this later time, meaning there is less demand for disease prevention methods. Evidently, the release of modified mosquitoes is a highly desirable technique, as even when the SIT programme costs ($C_3$) are roughly 100 times greater than the healthcare costs ($C_1$), a high release of modified mosquitoes is still of great benefit in managing dengue infections. This will largely be in part to the costs associated with the benefits of mosquito control outweighing the costs associated with the high numbers of infected hosts.

## 3.2. Case 2: both vaccination and vector control

We now include the potential for fully efficious vaccination ($u_1$) in the optimal control of dengue infections in a single settlement. Specific values for the daily cost of a vaccination programme are difficult to define precisely, however, Carrasco *et al.* [29] estimates the annual cost of control methods at \$50,000,000 ($\approx$ \$136,990 $d^{-1}$). We begin with the most idealized version of the vaccine, one which instantly moves susceptible hosts to the fully recovered class $R$ ($\xi = 1$), and which is fully effective, and able to be administered to the entire population. We fix the other costs, $C_1 = 510$ and $C_3 = 400$, and present our optimal solutions for varying vaccination costs ($C_2$) in figure 4.

Figure 4 shows that the optimum solution for all control costs is to provide the maximum amount of vaccine and SIT-mosquito coverage from as early as possible. These controls are slowly reduced to zero, reflecting how, as the host population moves from the susceptible to the recovered class by vaccine application, it becomes unnecessary to provide further vaccination. As the cost of controls increases, SIT and vaccination can be reduced at earlier points in time. $u_2$ is reduced to zero at earlier points in time as the cost, $C_2$, increases, however $u_1$ is always applied to some degree in the system until approximately day 22. Unsurprisingly, even when the cost of vaccination programmes ($C_2$) reaches 10 000 times the magnitude of vector control cost ($C_3$), the desirable impact of vaccination ensures it is always considered worthwhile. However, as the value of $C_2$ increases, we see the optimal recommended amount of $u_1$ begins to recede, while $u_2$ slowly increases. This result shows how the increasing cost of applying $u_1$ necessitates putting more emphasis on vector-targeted control methods to avoid an increase in the infected population. Of interest is that the optimal solution always involves a combination of both control methods. The rapid decline through time is owing to the vaccine

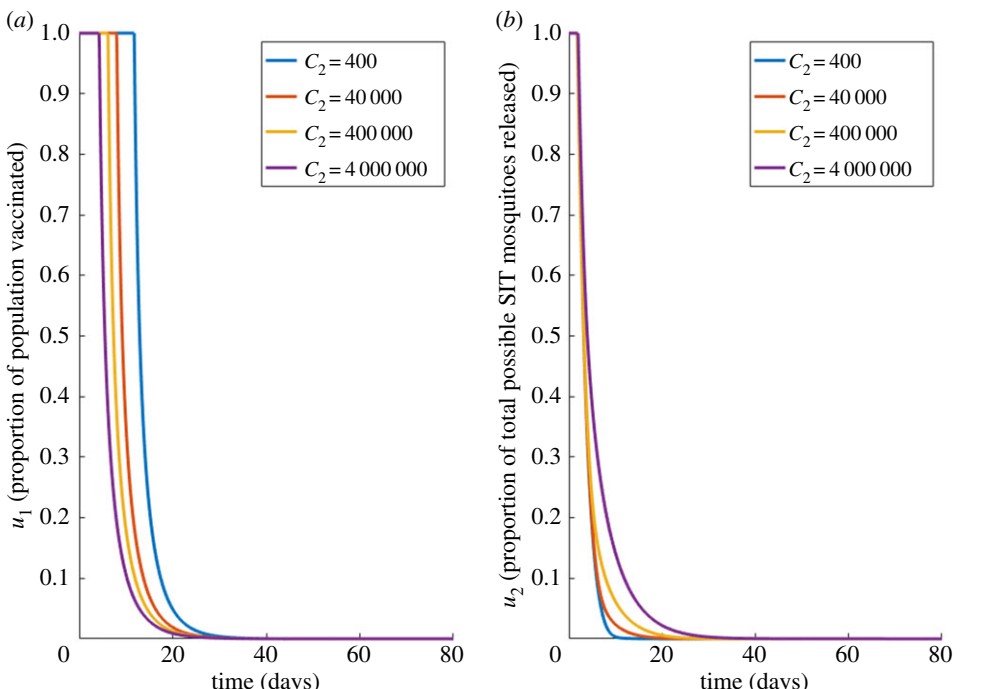

**Figure 4.** Case 2 results for fully efficacious vaccine. Optimal choices through time for; (a) the proportion vaccinated ($u_1$), (b) vector control ($u_2$), for different values of vaccination costs ($C_2$) for a fully efficacious vaccine ($\xi = 1$). Other costs held constant: $C_1 = 510$, $C_3 = 400$.

moving susceptible hosts to the recovered class, and hence reducing the potential for further infections, removing the need for further vaccination. To help understand the underlying dynamical impact, the electronic supplementary material, S4 appendix displays the dynamical behaviour of the underlying model state variables under this optimal scheme.

We repeat this analysis to investigate when the vaccine is imperfect ($\xi = 0$). Imperfect vaccination moves susceptible hosts to $R^B$ instead of $R$. These results are shown in figure 5.

Even though the vaccine is imperfect, vaccine delivery remains a feasible way to control dengue infection (figure 5a). However, coupling this with SIT vector control, can further reduce the impact from the primary and secondary dengue infections caused by the imperfect vaccine strategy (figure 5b). Interestingly, there is a single optimal solution to vector control releases for all vaccination costs under imperfect vaccines, this is owing to the new, limited effectiveness of the vaccine. Because all people will be at risk of a secondary infection, vector control methods will always be required in equal measure to tackle this threat, regardless of how many vaccines are administered.

To explore this interplay between vaccination and vector control further, we investigate restrictions in vaccine coverage. It is unrealistic to assume a vaccine can be applied to the entire population simultaneously. For this reason, we introduce a new constant ($\sigma = 0.5$), to describe reduced effectiveness of vaccine coverage. Linked to this, Hadinegoro et al. [30] have suggested that Dengvaxia may lead to imperfect resistance and efficacy. $\xi$ is kept at 0 as well for this case. The governing equations (and associated adjoints) for $S$, $R^B$ and $R$ are now,

$$\frac{\mathrm{d}S}{\mathrm{d}t} = -\frac{Sab_1(X+Y)}{N} - \sigma u_1 S,$$

$$\frac{\mathrm{d}R^B}{\mathrm{d}t} = \rho_1 I^B - \eta R^B + \sigma(1-\xi)u_1 S$$

and

$$\frac{\mathrm{d}R}{\mathrm{d}t} = \rho_2(I^{B_2} + I^{A_2}) + \sigma \xi u_1 S.$$

Figure 6 displays the optimal controls for a variety of vaccine costs ($C_2$).

Weaker coverage slightly reduces the effectiveness of the vaccine. However, a prolonged application of both controls at the highest possible level remains the optimal solution (figure 6). Again, the single line in figure 6b represents the single best control $u_2$ for all varying vaccination costs.

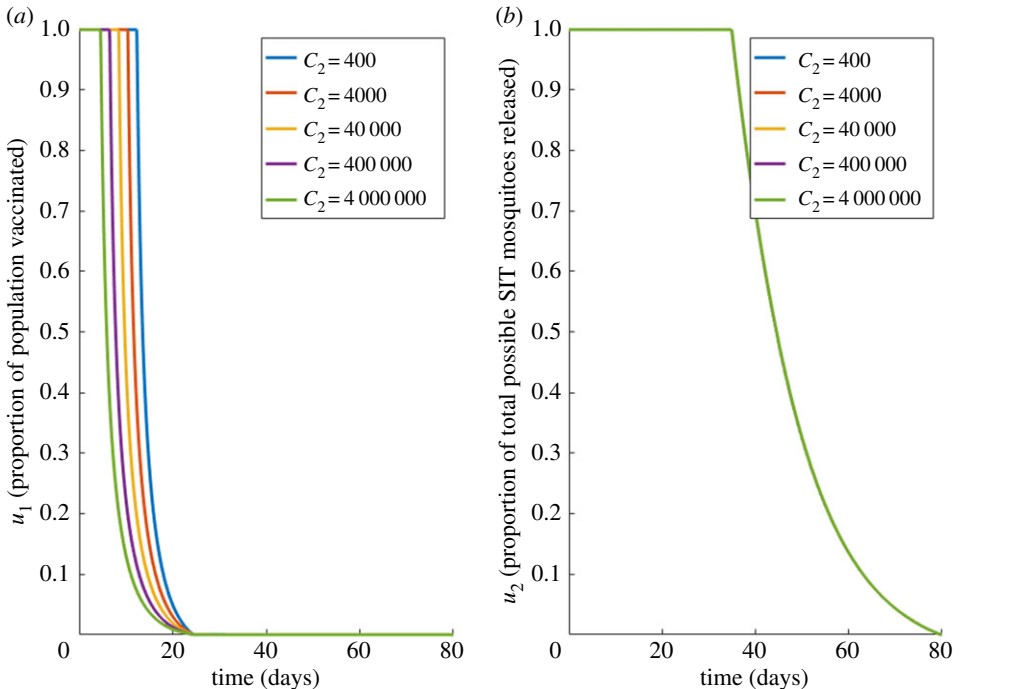

**Figure 5.** Case 2 results for imperfect vaccine. Optimal choices through time for; (*a*) the proportion vaccinated ($u_1$), (*b*) vector control ($u_2$), for different values of vaccination costs ($C_2$) under imperfect vaccination ($\xi = 0$). Other costs held constant: $C_1 = 510$, $C_3 = 400$.

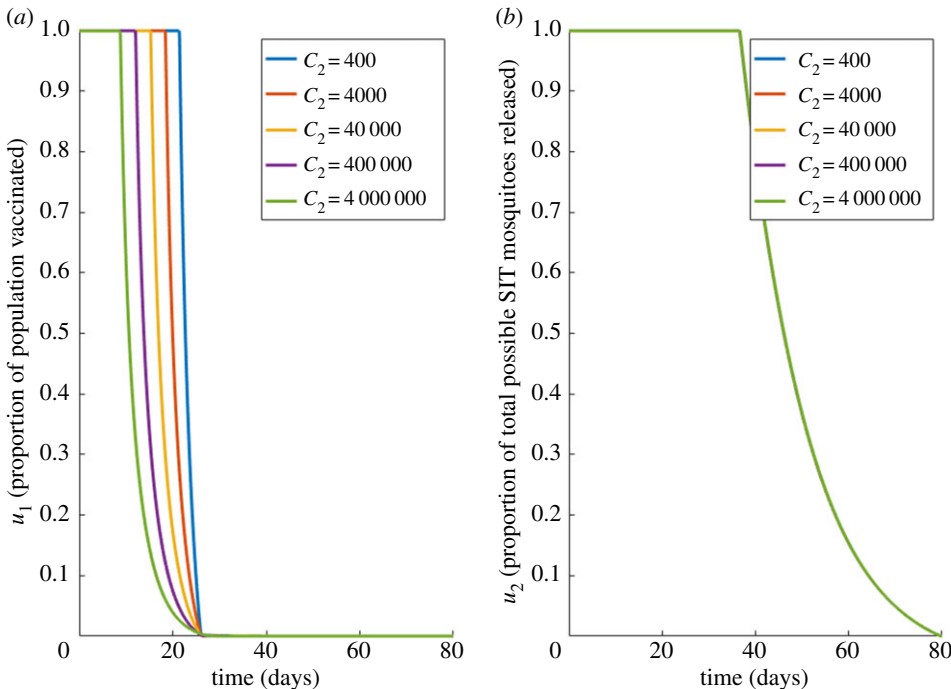

**Figure 6.** Case 2 results for imperfect vaccine with reduced coverage. Optimal choices through time for; (*a*) the proportion vaccinated ($u_1$), (*b*) vector control ($u_2$), for different values of vaccination costs ($C_2$) under imperfect vaccine ($\xi = 0$) and reduced vaccine coverage ($\sigma = 0.5$). Other costs held constant: $C_1 = 510$, $C_3 = 400$.

### 3.3. Case 3: high initial dengue infection

The first two cases have dealt with epidemic cases, where our system started with just two infected hosts, and only 4000 infected vectors. We now consider disease scenarios with far larger initial infected populations. For this case scenario, we keep our canonical choices of $C_1 = 510$, $C_3 = 400$, $\xi = 0$ and $\sigma =$

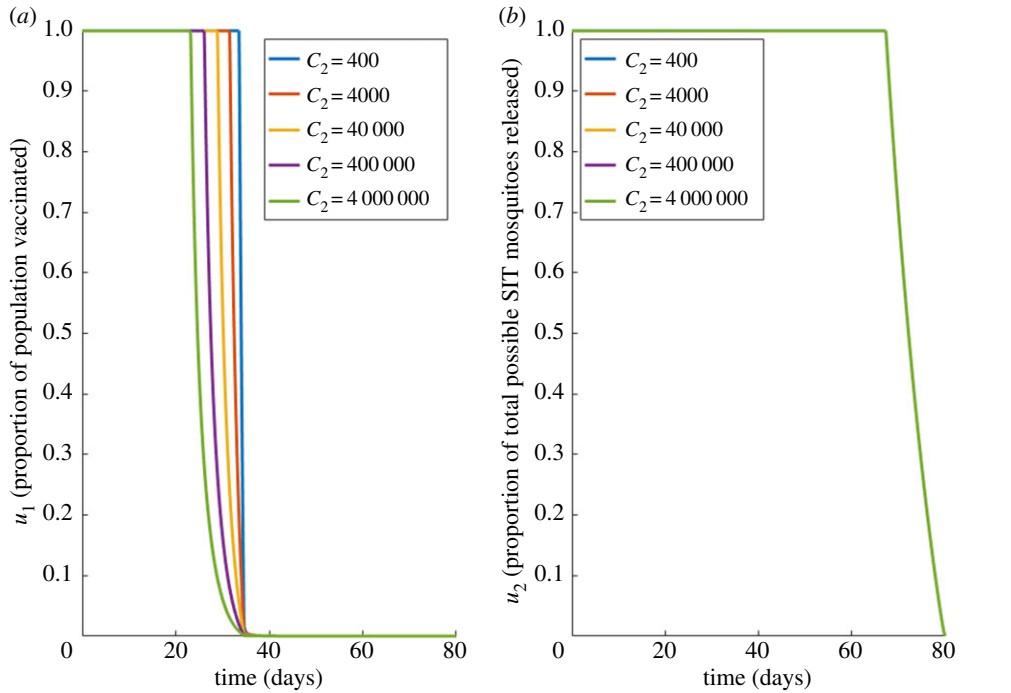

**Figure 7.** Case 3 results. Optimal choices through time for; (a) the proportion vaccinated ($u_1$), (b) vector control ($u_2$), for different values of vaccination costs ($C_2$) with high initial dengue infections. Other costs held constant: $C_1 = 510$, $C_3 = 400$, $\xi = 0$ and $\sigma = 0.5$. This is for an example of high initial infections, where $I_0^A = I_0^B = 1\,000\,000$.

0.5, but we increase our initial value of infected hosts to $I_0^A = 1\,000\,000$ and $I_0^B = 1\,000\,000$. Figure 7 shows our optimal results for a variety of choices of $C_2$.

In cases of high infected populations, there is clearly a greater need for integrated control efforts. Both vaccination and vector control need to be sustained longer to achieve control. A rapid decline in vaccination is then expected as the population of susceptible hosts, $S$, is moved to a recovered class hence reducing the need for further vaccination (figure 7a). To maintain the disease control, high levels of modified mosquito releases are needed to ensure the control is cost-effective (figure 7b). The constraint(s) of cost-effective disease control within a feasible time frame (e.g. $t = 0$ days $\rightarrow t = 80$ days) is commonly seen in our optimal control results, where controls may be sharply curtailed at the end, in an effort to reduce cost before the impact on the number of infected can be realized.

## 3.4. Case 4: multiple settlements

Finally, we consider a case with inter-settlement dynamics under epidemic dengue dynamics ($I_0^A = I_0^B = 1$). We consider a perfectly acting vaccine that provides immunity to only one serotype ($\xi = 0$, $\sigma = 1$). All towns start with the same initial conditions of $S_0^i = 10\,000$, $J_0^i = 400\,000$ and $M_0^i = 20\,000$ for $i \in \{2, 3, \dots, n\}$. All other town states begin at 0. Note that the towns begin with no infection, either in their hosts or vectors. Costs are set at similar values to above ($C_1 = 510$, $C_2 = 4000$ and $C_3 = 400$).

Figure 8a,b display the optimal control variables within the central city for an increasing network size and total settlement numbers, $n$. Figure 8c,d meanwhile display the optimal control variables for the town settlements for the same varieties of $n$. Similar to the previous cases, vaccination ($u_1$) is optimally administered at the maximum level at the beginning of the disease outbreak, before eventually being reduced to zero by approximately day 22. While this is optimal for both the central hub and the outer towns, coverage is reduced at a slower rate from an earlier point in time than for the central city. In the case of SIT-mosquito releases ($u_2$), these modified insects are optimally applied in high amounts within the central cities, but only released at very small levels in the outer towns. Interestingly, within the central city the optimal control strategies are the same regardless of the number of small towns. This is probably owing to the relative size of the towns compared to the city. As only a very small percentage of hosts move from each town this has very little impact on the total epidemiological dynamics. Note that the optimal result displayed is the same for each individual town.

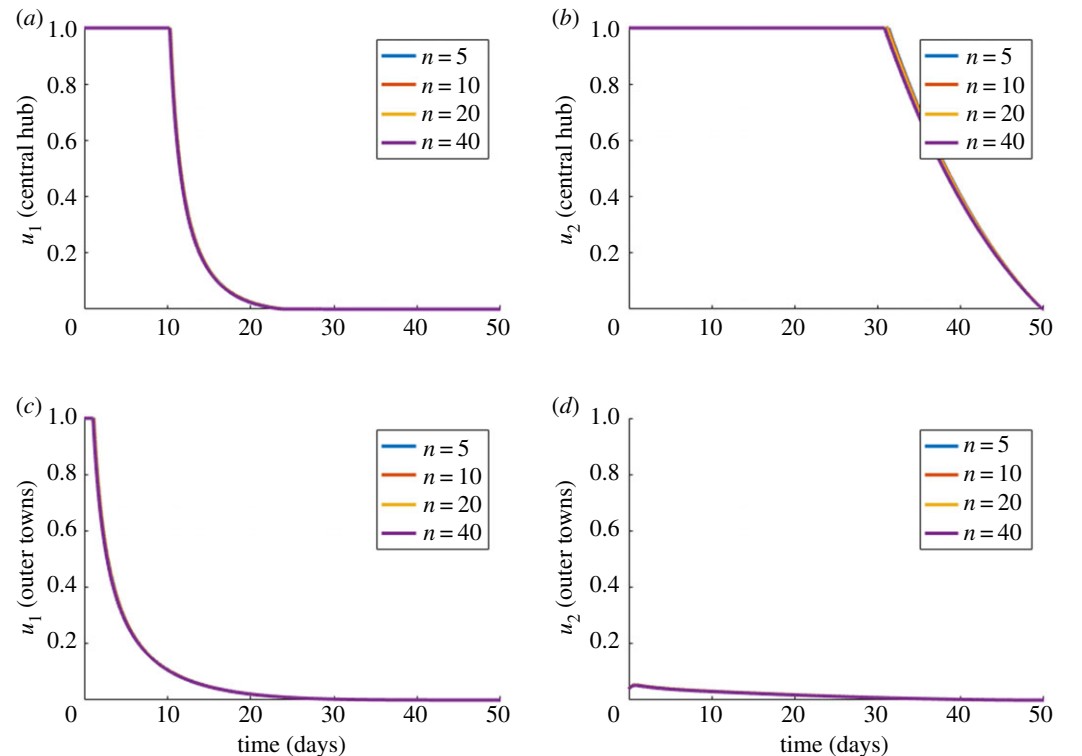

**Figure 8.** Case 4 results for varying $n$. Optimal choices through time for different network sizes ($n$). (a) Proportion vaccinated ($u_1$) in the central city settlement. (b) Vector control ($u_2$) in the central city settlement. (c) Proportion vaccinated ($u_1$) in the outer town settlements. (d) Vector control ($u_2$) in the outer town settlements. Vaccine deployment is expected to decline owing to reductions in the number of susceptibles. Optimal control regimes are identical in every outer town settlement.

The same optimal control results are recommended for each town, regardless of how many towns there may be in the network. This is linked directly to the relatively small impact the number of towns had on the overall dynamics within the city (as seen in figures 8a,b). Adding an additional spoke settlement tends not to affect the central city, which is the only way towns could ever be seen to 'communicate'. However, the form of control in towns is different to that recommended for the central city. Relatively low levels of coverage and declines in vaccination and vector control (after an initial increase) are the optimal strategy. It seems optimal control methods for the spokes and hub remain somewhat independent.

A final sensitivity analysis was made to observe the effects of varying the rate of migration of susceptible hosts ($m_S$) on control outcomes. All other constants were kept as initially stated for case 4 and we chose $n = 5$ as a fixed number of settlements. These results are shown in figure 9. Changing the value of $m_S$ had no impact on the control scheme for the city (figure 9a,b), however, movement of susceptible hosts had a dramatic influence on the optimal controls for the towns themselves (figure 9c,d). This is to be expected, as these outer towns are initialized with no infected hosts or vectors. As such, the settlement may only become infected owing to the migration effect, with hosts becoming infected in the central node, and then returning to the outer node to infect the vector population. This greater level of migration results in a more erratic optimal $u_1$ for the outer towns (figure 9c) owing to the rapidly changing influxes of infected individuals owing to a greater migration constant. Note also how recommended increases to $u_1$ and $u_2$ gradually decrease as $m_S$ increases in value.

## 4. Discussion

Here, we have investigated how vaccination and SIT-focused vector control approaches can be best combined to manage dengue disease dynamics in small networks. Using optimal control approaches, we have shown that a combination of control measures is always an optimal choice, regardless of all the factors explored in the above case studies. This key finding corroborates that presented by Hendron & Bonsall [16], whose work highlighted the synergistic relationship between vaccine

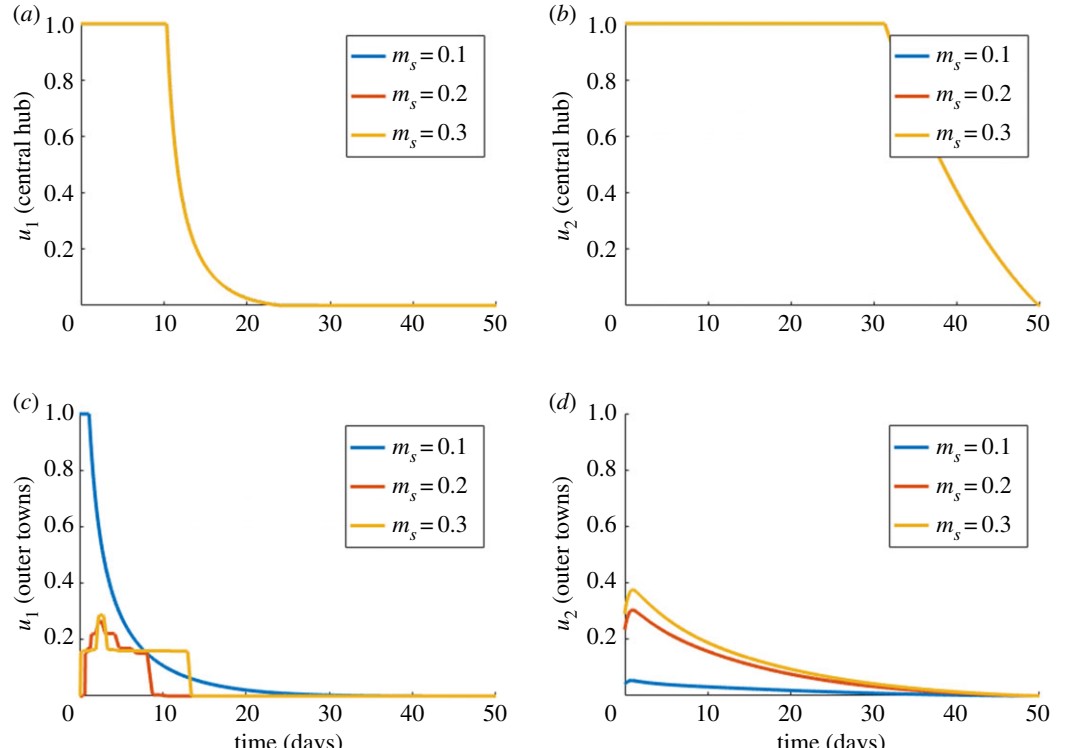

**Figure 9.** Case 4 results for varying $m_S$. Optimal choices through time for; (a) proportion vaccinated ($u_1$) in a central hub settlement, (b) vector control ($u_2$) in a central hub settlement, (c) proportion vaccinated ($u_1$) in an outer town settlement, (d) vector control ($u_2$) in an outer town settlement, for different values of host movement ($m_S$). Here, $n = 5$ and costs were set at $C_1 = 510$, $C_2 = 4000$ and $C_3 = 400$.

coverage and vector control across small networks. Our optimal control approach has further elucidated this relationship and demonstrated how these treatments can be most effectively combined and implemented to achieve cost-effective control. Furthermore, we have revealed how unique factors of an outbreak can alter the optimal course of action.

A key finding, throughout all the scenarios we investigated, is how minimizing the infected population is always of the greatest economic importance. As shown in figure 3, even when increasing the cost of vector control to a far higher value than is realistic, any control method is always financially advantageous. While calculating specific hospitalization costs is difficult, an in-depth review by Stahl *et al.* [31] found that countries which invested greatly in vector control methods had lower total costs owing to dengue *per capita*. For some countries, treatment costs are still greater than vector control costs, and this presents a significant financial burden [32].

Our results have also shown that a combination of prevention methods is always preferred, even when there is considerable disparity between their respective costs (figure 4). Other studies on the control of dengue transmission draw similar conclusions (e.g. [33]). This disparity in costs is best explained from our analysis of imperfect vaccines (figure 5). The two methods target two different frameworks; hosts and vectors. As these methods affect separate targets, the impact of one on the model cannot be perfectly offset by the other. When an imperfect vaccine was considered, the optimal application of modified mosquito release methods remained the same for all vaccine costs, owing to the vaccine being unable to transfer hosts beyond the risk of secondary infection (hosts were moved to $R^B$ instead of $R$), hence vector control methods were always required to mitigate this consistent flux of susceptibles. A recent study showed similar results; Fitzpatrick *et al.* [34], using a dynamic Markov approach, showed that vector control remains a cost-effective treatment strategy even in the presence of a medium-efficacy vaccine. Together with our findings, this suggests that vector control schemes should always be employed as a safeguard against imperfect vaccines.

Case 3, an instance of a higher initially infected population ($I_0^A = 1\,000\,000$ and $I_0^B = 1\,000\,000$), further strengthens the argument for a strong vector control programme regardless of vaccine availability. When a high initial disease burden was considered ($S_0 = 2\,000\,000$, $I_0^A = 1\,000\,000$, $I_0^B = 1\,000\,000$), a consistent maximum application of modified mosquito release was considered optimal, as well as a strong initial

supply of vaccines. The epidemiological reasoning behind this is that the two million hosts that begin the case study with a primary infection, will never be able to benefit from a vaccine, as the framework does not provide them with an option to then receive such treatment. Biologically, this can be interpreted by drawing parallels from cost-effective inventions to control epidemic and endemic cholera [35]. While cholera vaccines proved very cost-effective treatments in small epidemic outbreaks, under endemic parameters vaccines were ineffective and it was considered more cost-effective to focus on the provision of safe drinking water and clearing the surrounding environment of the bacteria. While the epidemiology is different, the phenomena is the same, where vaccines are no longer capable of providing a herd immunity effect [36] alternative solutions to mitigate disease burden need to be sought.

One key result (in case 4) is where control schemes in the main city were unaffected by changes to the nature of the surrounding towns (figure 8). This result suggests that city-wide control measures may not need to factor migration into their disease-prevention programmes. A programme that effectively aids the city will effectively aid those that migrate in by definition, even when not specifically vaccinating those that live in the outer towns. It confirms that focusing efforts on the largest population centres is by far the best strategy. This is in agreement with the general theoretical findings that, when vaccine supply is limited, they are best distributed among the groups who are most crucial to transmission [37]. To aid outer towns further, our results suggest a very small-scale vaccination programme and SIT mosquito release within the towns themselves would be appropriate. This would probably just strengthen a herd-immunity effect. By far the biggest impact on minimizing outbreaks in outer towns however is ensuring the central cities are kept under control to minimize the chances of the disease even reaching these towns. We do, however, see that in cases of higher migration (figure 9d), a longer sustained town-based SIT mosquito release plan is suggested to help protect the town itself.

While an optimal control was quickly found for the city settlements, an optimal solution within the outer towns was more complicated. When the migration rate $m_S$ was increased, the method took longer to converge and displayed more unpredictable results (figure 9c). This result suggests that the most effective way to control disease outbreaks in the towns was an effective city-based control scheme. The application of vector control interventions in the towns was less efficacious, and therefore no clear 'optimal solution' could easily be found. Also, the fact that the optimal controls in the towns were of such small value, their input into the cost function was somewhat negligible. This explains an undocumented phenomena where convergence was more easily acquired for large cost values of $C_2$ and $C_3$. This warrants further investigation and suggests it could be wise to re-consider the formulation of the cost function, perhaps changing the controls to represent a *volume* of application as opposed to a *proportion* $(0 \le u_1/u_2 \le 1)$.

While the specific values of the control variables may have varied for different case studies, all of these supported a strong initial programme of vaccines (to push as many hosts into the recovered class as quickly as possible) followed by modified mosquito release (which in turn suppress the number of infected mosquitoes through the season). Often modified mosquito releases need to be sustained longer than vaccination programmes to achieve disease control. Although dengue vaccines can reduce susceptible numbers, mosquito control acts to suppress population sizes below the key entomological threshold to prevent further primary and secondary infections emerging. This critical combination of control interventions achieves dengue suppression most effectively in all cases.

## 4.1. Coda

The optimal control problem of combining vector interventions and vaccination has yielded key insights into dengue disease management. However, a number of caveats require more careful consideration.

As is always the case, our mathematical model is a major simplification of biological reality, and several steps could be taken to improve the model. We note that our cost function could be greatly improved. As well as the ideas posed above, as it stands, the cost of fully vaccinating a small town is the same per day as vaccinating the large city under our model. This is a clearly a convenient fiction and requires further investigation. Likewise, the cost of SIT programmes is complicated by the fact that an initial set-up cost is always required to build the breeding facility. Many of our optimal results eventually push the vector control $(u_2)$ to zero to remove all cost. However, the averaged 'daily cost' of having built the facility needs to be considered. The nature of the model also urges an optimal result to maximize flow of hosts into the fully recovered class, $R$, whereas it may be more realistic to consider a delay, where recovered individuals may be temporarily fully immune $(R)$ before rejoining one of the susceptible pools $(S, S^{A_2}$ or $S^{B_2})$ and then being exposed to a different strain of dengue.

The limit on migration only occurring into the central city was an important step for formulating a simple network model of dengue disease dynamics. Indeed, our case studies will only represent certain particular cases of dengue epidemics. Many dengue-endemic areas can be characterized by recurring outbreaks, while our current formulation focuses on only one contained outbreak. Considering inter-town dynamics and travel between smaller towns and villages, could give key insight on how to combat the disease across all settlements outside of cities. Several more key scenarios can be considered, for example, larger settlements with a large host population, and high vector populations, would pose a significant challenge.

From a computational perspective, the convergence and efficiency of optimal control methods is very sensitive to certain parameters. Run-time was significantly reduced when generating a new expression for $u$ by choosing a weighting of $u = (0.7 * u_{\text{old}} + 0.3 * u_{\text{new}})$, to save time lost owing to 'over-shooting'. Because this in an iterative procedure, converging to the optimal $u$, using the more standard update criteria of $u = (0.5 * u_{\text{old}} + 0.5 * u_{\text{new}})$ was found to repeatedly overcompensate in this instance, by making unnecessarily large adjustments to $u_{\text{old}}$. Developing aspects of Pontryagin's maximum principle for networks is clearly in its infancy and there is much to be done in developing the mathematics for approaching these challenging real-world problems.

All the caveats notwithstanding, the optimal control approach and insights developed here provides a key tool in developing a richer understanding of how to combine integrated vector-control programmes in managing the public health burden of dengue.

Data accessibility. All code used in calculating results and generating the presented figures is available at; https://osf.io/vspxe/ [28].

Authors' contributions. M.B.B. conceived and laid out the goals and methodology. T.R. built the required optimal control framework. T.R. and K.E.W. performed the numerical simulations. T.R. wrote the paper. M.B.B. and K.E.W. helped revise the paper.

Competing interests. We declare we have no competing interests.

Funding. The work was supported through an Engineering and Physical Sciences Research Council (EPSRC) (https://epsrc.ukri.org/) Systems Biology studentship award (EP/G03706X/1 to T.R.) and a Defense Advanced Research Projects Agency (DARPA) (www.darpa.mil/) grant no. (HR0011-16-2-0006 to M.B.B.). The funders had no role in study design, data collection and analysis, decision to publish, or preparation of the manuscript.

Acknowledgements. We thank the Mathematical Ecology Research Group in Oxford and Angela McLean for discussion and comments on the ideas presented in this work.

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
