## [Reviewer comments · Royal Society Open Science]

Review History

RSOS-181843.R0 (Original submission)

Review form: Reviewer 1

Is the manuscript scientifically sound in its present form?

Yes

Are the interpretations and conclusions justified by the results?

Yes

Is the language acceptable?

Yes

Is it clear how to access all supporting data?

Yes

Do you have any ethical concerns with this paper?

No

Have you any concerns about statistical analyses in this paper?

No

Recommendation?

Major revision is needed (please make suggestions in comments)

Comments to the Author(s)

This manuscript takes the model from Hendron and Bonsall 2016 on “The interplay of vaccination and vector control on small dengue networks” and uses optimal control theory to address the same questions. This manuscript appears to come up with many of the same conclusions as the earlier paper.

My main issue with the paper was that neither the Hendron and Bonsall model, nor optimal control theory, were explained quite well enough in this manuscript to quite understand all of the details.

In particular, in the model (equations 1-9), it’s difficult to understand what is being assumed in all of the terms involving the symbol capital Omega, which is meant to be the total net movement of hosts into and out of settlement i . More explanation is needed here (e.g. why is it $(N_i + \text{Omega}_i)$ in the denominator?)

Similarly, although it may be explained in the Hendron and Bonsall paper, I really couldn’t figure out what the $(Z_i / (Z_i + \psi_i * u_2))$ terms in equation 11 were assuming for the juvenile mosquitoes.

Also in Equation 11, why is the form $J_1 * \ln(1 + \delta * J_1)$ used for the density-dependence term? That seems like a rather unusual form for intraspecific competition.

Line 154: Optimal control: Appendix S1 gives a very brief, and general, explanation for optimal control, but it seems that few more details should go into the main text, in order for the readers to be able to understand it. For example, in the paragraph starting on line 171, the term adjoint functions is not defined, and it is not explained that g , the governing ODEs, are simply the model defined above.

Line 165: The explanation about using the square of the variables in the cost function is less than convincing. It simply says that it avoids incorrectly assuming a linear relationship. But, could it instead be incorrectly assuming a quadratic relationship (without a linear term)?

Line 184: It seems like a very strange assumption to have people travel to the central settlement, but never travel back again to their home settlement.

Figures 3-9: Need more explanation on how to interpret the figures.

For example, is Figure 3 just saying that it’s useful to perform vector control, releasing all possible SIT modified mosquitoes, up to some time point (50 to 80 days in the outbreak, depending on the cost), and then not so much? But, can we tell from this figure how helpful vector control actually is from the figure?

In order to interpret the effect of the timing of the release, would it be useful to also show what’s happening to the disease outbreak, i.e. to show a trajectory of the population dynamics from the model to show what’s happening on day 50ish of the outbreak, with and without control?

It also might be useful to label the y-axis not just u_2 , but also include the definition of u_2 “Proportion of total possible SIT modified mosquitoes released”, so that it can be instantly interpreted. (I have the same comment for all of the subsequent figures).

line 207 states "Evidently, the release of modified mosquitoes is a powerful technique". How can we tell this from the figure? The figure just shows that the optimal strategy is to use the modified mosquitoes, but it doesn't show how much it actually reduces the disease outbreak.

Line 313: I'm confused by the statement "as migration is the sole source of infection for these outer towns". How does this work? I thought that movement was only from the towns to the central city.

Line 440: This paragraph is hard to understand without more details of how the optimal control optimization works.

Throughout the manuscript: A canonical set of parameters is used throughout. Some of these parameters have a surprising level of precision (e.g. 0.0541). How much do the conclusions of the manuscript depend on the exact choice of these parameters?

Minor issues:

Abstract: There's an extra "that" in the sentence: "We show that that a combination..."

Line 430: extra ", " in: "before rejoining ,one"

Decision letter (RSOS-181843.R0)

24-Jun-2019

Dear Mr Rawson,

The editors assigned to your paper ("Optimal control approaches for combining medicines and mosquito control in tackling dengue") have now received comments from reviewers. We would like you to revise your paper in accordance with the referee and Associate Editor suggestions which can be found below (not including confidential reports to the Editor). Please note this decision does not guarantee eventual acceptance.

Please submit a copy of your revised paper before 17-Jul-2019. Please note that the revision deadline will expire at 00.00am on this date. If we do not hear from you within this time then it will be assumed that the paper has been withdrawn. In exceptional circumstances, extensions may be possible if agreed with the Editorial Office in advance. We do not allow multiple rounds of revision so we urge you to make every effort to fully address all of the comments at this stage. If deemed necessary by the Editors, your manuscript will be sent back to one or more of the original reviewers for assessment. If the original reviewers are not available, we may invite new reviewers.

- Data accessibility

If you wish to submit your supporting data or code to Dryad (<http://datadryad.org/>), or modify your current submission to dryad, please use the following link:
<http://datadryad.org/submit?journalID=RSOS&manu=RSOS-181843>

- Competing interests

- Authors' contributions

- Acknowledgements

- Funding statement

on behalf of Mark Chaplain (Subject Editor)
 openscience@royalsociety.org

Associate Editor Comments to Authors:

Firstly, please accept our sincere apologies for the delay to the final decision for your paper. As you can see, we unfortunately experienced some difficulty securing the desired number of reviewers for your work, however in order to prevent any further delay, the Editor felt that it would be best to send your decision onto you now.

We would be grateful if you could revise your manuscript in accordance with the reviewer's comments below, and then resubmit your paper for consideration. Please ensure that you provide a point-by-point response file to the reviewer's concerns when resubmitting your paper. Additionally, please note that upon resubmission, a second reviewer will likely be sought at the Editor's discretion.

We hope that the following comments are useful for your revision.

Comments to Author:

Reviewers' Comments to Author:
 Reviewer: 1

Comments to the Author(s)

This manuscript takes the model from Hendron and Bonsall 2016 on "The interplay of vaccination and vector control on small dengue networks" and uses optimal control theory to address the same questions. This manuscript appears to come up with many of the same conclusions as the earlier paper.

My main issue with the paper was that neither the Hendron and Bonsall model, nor optimal control theory, were explained quite well enough in this manuscript to quite understand all of the details.

In particular, in the model (equations 1-9), it's difficult to understand what is being assumed in all of the terms involving the symbol capital Omega, which is meant to be the total net movement of hosts into and out of settlement i . More explanation is needed here (e.g. why is it $(N_i + \Omega_i)$ in the denominator?

Similarly, although it may be explained in the Hendron and Bonsall paper, I really couldn't figure out what the $(Z_i / (Z_i + \psi_i * u_2))$ terms in equation 11 were assuming for the juvenile mosquitoes.

Also in Equation 11, why is the form $J_{i-1} * \ln(1 + \delta * J_{i-1})$ used for the density-dependence term? That seems like a rather unusual form for intraspecific competition.

Line 154: Optimal control: Appendix S1 gives a very brief, and general, explanation for optimal control, but it seems that few more details should go into the main text, in order for the readers to be able to understand it. For example, in the paragraph starting on line 171, the term adjoint functions is not defined, and it is not explained that g , the governing ODEs, are simply the model defined above.

Line 165: The explanation about using the square of the variables in the cost function is less than convincing. It simply says that it avoids incorrectly assuming a linear relationship. But, could it instead be incorrectly assuming a quadratic relationship (without a linear term)?

Line 184: It seems like a very strange assumption to have people travel to the central settlement, but never travel back again to their home settlement.

Figures 3-9: Need more explanation on how to interpret the figures.

For example, is Figure 3 just saying that it's useful to perform vector control, releasing all possible SIT modified mosquitoes, up to some time point (50 to 80 days in the outbreak, depending on the cost), and then not so much? But, can we tell from this figure how helpful vector control actually is from the figure?

In order to interpret the effect of the timing of the release, would it be useful to also show what's happening to the disease outbreak, i.e. to show a trajectory of the population dynamics from the model to show what's happening on day 50ish of the outbreak, with and without control?

It also might be useful to label the y-axis not just u_2 , but also include the definition of u_2 "Proportion of total possible SIT modified mosquitoes released", so that it can be instantly interpreted. (I have the same comment for all of the subsequent figures).

line 207 states "Evidently, the release of modified mosquitoes is a powerful technique". How can we tell this from the figure? The figure just shows that the optimal strategy is to use the modified mosquitoes, but it doesn't show how much it actually reduces the disease outbreak.

Line 313: I'm confused by the statement "as migration is the sole source of infection for these outer towns". How does this work? I thought that movement was only from the towns to the central city.

Line 440: This paragraph is hard to understand without more details of how the optimal control optimization works.

Throughout the manuscript: A canonical set of parameters is used throughout. Some of these parameters have a surprising level of precision (e.g. 0.0541). How much do the conclusions of the manuscript depend on the exact choice of these parameters?

Minor issues:

Abstract: There's an extra "that" in the sentence: "We show that that a combination..."

Line 430: extra ",," in: "before rejoining ,one"

Author's Response to Decision Letter for (RSOS-181843.R0)

See Appendix A.

RSOS-181843.R1 (Revision)

Review form: Reviewer 2

Is the manuscript scientifically sound in its present form?

No

Are the interpretations and conclusions justified by the results?

No

Is the language acceptable?

Yes

Do you have any ethical concerns with this paper?

No

Have you any concerns about statistical analyses in this paper?

No

Recommendation?

Major revision is needed (please make suggestions in comments)

Comments to the Author(s)

In this paper, Rawson et al. use optimal control theory on a previously published dengue model, in order to identify the best combination of vaccination and sterile insect release to reduce the number of infections.

I recognise the interest and usefulness of the optimal control methodology for epidemiological models; however, I have a series of major concerns regarding the hypotheses on dengue dynamics in both the model and the objective function, which may affect the conclusions, especially those formulated in terms of policy recommendations.

Major comments.

1. The model aims to minimize the total number of infections and associates a hospitalization cost to each infection. In dengue epidemics, there is potentially a very large proportion of asymptomatic or mildly symptomatic cases, that would not require hospitalization. Unless the current model parameterization represent the dynamics of severe cases only, this hypothesis in the cost function may not be realistic for dengue.
2. Surprisingly, disease induced mortality is included in this model without demography (does the human population stay constant?), but not in the cost function, giving the impression that deaths are not accounted for in the optimization of intervention strategies.
3. The time indices over which the objective function is integrated (0 and t_f) are not clearly defined: because they represent the period of time over which costs are calculated, they may impact the results of the optimization and should therefore be detailed and discussed. In addition, I do not understand what the time dimension in the figures refers to (introduction of interventions? Duration of intervention? Time period considered in the objective function?).
4. The hypotheses on vaccine effect do not seem to represent the current Dengvaxia vaccine, which, to my knowledge, is only recommended for individuals with pre-existing dengue immunity, and not for naïve individuals (S).
5. The initialisation of the model in a naïve population ($R=0, S_a2=0, S_b2=0, R_a=0, R_b=0$) does not reflect current recommendation for the vaccine, which would not be administered in a naïve population, but in a population with high endemicity. In addition, the text mentions that the dynamic of the model consists in one "disease outbreak" (L256): does this mean the studied trajectory does not include oscillations or endemicity? A graphic representing the disease dynamics (e.g. number infected individuals over time) should be included to clarify this.

6. I understand that the main objective of the manuscript is not to perform a sensitivity analysis; however, the model relies on many parameters and assumptions that are neither varied nor even discussed. This is even more important for some parameters which, to my knowledge, are not precisely quantified and still debated in the literature (duration of cross-immunity, some mosquito parameters, people's movements)

Minor comments

1. The writing of the equations could be simplified to ease the reading (merge the sums, simplify the notations for parameters which take the same value throughout the paper (e.g. ρ_1 and ρ_2), change the notation g which is used twice)
2. Table 1 could be more precise in giving the reference for the value of each parameter
3. L13: Five dengue serotypes ?
4. The conclusions refer sometimes broadly to "vector control" whereas the model only considers the SIT intervention (or genetic based methods similar to it) which is a rather new and specific intervention compared to vector control in general.
5. I would suggest to include some literature on the use of optimal control theory for epidemiological models in the main text (and not only the supplement). Some more details on the technicalities of the method which are quickly introduced in the conclusions (such as weighting, sensitivity) could also be added (maybe in the supplement) to better inform readers interested in this methodology.
6. In the code provided online: could you explain why two separate scripts are available?

Decision letter (RSOS-181843.R1)

14-Jan-2020

Dear Mr Rawson:

Manuscript ID RSOS-181843.R1 entitled "Optimal control approaches for combining medicines and mosquito control in tackling dengue" which you submitted to Royal Society Open Science, has been reviewed. The comments of the reviewer(s) are included at the bottom of this letter.

Please submit a copy of your revised paper before 06-Feb-2020. Please note that the revision deadline will expire at 00.00am on this date. If we do not hear from you within this time then it will be assumed that the paper has been withdrawn. In exceptional circumstances, extensions may be possible if agreed with the Editorial Office in advance. We do not allow multiple rounds of revision so we urge you to make every effort to fully address all of the comments at this stage. If deemed necessary by the Editors, your manuscript will be sent back to one or more of the original reviewers for assessment. If the original reviewers are not available we may invite new reviewers.

- Ethics statement

- Data accessibility

- Competing interests

- Authors' contributions

- Acknowledgements

- Funding statement

Kind regards,
Andrew Dunn
Royal Society Open Science Editorial Office

on behalf of Prof Mark Chaplain (Subject Editor)
 openscience@royalsociety.org

Associate Editor Comments to Author:

Thank you for your patience with us: unfortunately, the reviewers of the original submission were not available to assist with this revision, and it took an unusual length of time (and large number of invitations) to solicit alternative commentary.

Generally, Royal Society Open Science does not permit multiple rounds of revision, but given the difficulty in securing review, and recognising that a number of these comments may be new, we'd like to offer you the opportunity to revise the manuscript to take into account this new referee's comments - they will be invited to assess your changes, and we'd strongly recommend you prepare not only a tracked-changes version of the revision but also a clear point-by-point response to their recommendations. If you need any additional time to complete the revisions, do let the editorial office know.

Thanks again for your patience and we look forward to receiving your revision in due course.

Reviewer comments to Author:

Reviewer: 2

Comments to the Author(s)

In this paper, Rawson et al. use optimal control theory on a previously published dengue model, in order to identify the best combination of vaccination and sterile insect release to reduce the number of infections.

I recognise the interest and usefulness of the optimal control methodology for epidemiological models; however, I have a series of major concerns regarding the hypotheses on dengue dynamics in both the model and the objective function, which may affect the conclusions, especially those formulated in terms of policy recommendations.

Major comments.

1. The model aims to minimize the total number of infections and associates a hospitalization cost to each infection. In dengue epidemics, there is potentially a very large proportion of asymptomatic or mildly symptomatic cases, that would not require hospitalization. Unless the current model parameterization represent the dynamics of severe cases only, this hypothesis in the cost function may not be realistic for dengue.
2. Surprisingly, disease induced mortality is included in this model without demography (does the human population stay constant?), but not in the cost function, giving the impression that deaths are not accounted for in the optimization of intervention strategies.
3. The time indices over which the objective function is integrated (0 and t_f) are not clearly defined: because they represent the period of time over which costs are calculated, they may impact the results of the optimization and should therefore be detailed and discussed. In addition, I do not understand what the time dimension in the figures refers to (introduction of interventions? Duration of intervention? Time period considered in the objective function?).
4. The hypotheses on vaccine effect do not seem to represent the current Dengvaxia vaccine, which, to my knowledge, is only recommended for individuals with pre-existing dengue immunity, and not for naïve individuals (S).
5. The initialisation of the model in a naïve population ($R=0$, $Sa_2=0$, $Sb_2=0$, $Ra=0$, $Rb=0$) does not reflect current recommendation for the vaccine, which would not be administered in a naïve population, but in a population with high endemicity. In addition, the text mentions that the dynamic of the model consists in one "disease outbreak" (L256): does this mean the studied

trajectory does not include oscillations or endemicity? A graphic representing the disease dynamics (e.g. number infected individuals over time) should be included to clarify this.

6. I understand that the main objective of the manuscript is not to perform a sensitivity analysis; however, the model relies on many parameters and assumptions that are neither varied nor even discussed. This is even more important for some parameters which, to my knowledge, are not precisely quantified and still debated in the literature (duration of cross-immunity, some mosquito parameters, people's movements)

Minor comments

1. The writing of the equations could be simplified to ease the reading (merge the sums, simplify the notations for parameters which take the same value throughout the paper (e.g. ρ_1 and ρ_2), change the notation g which is used twice)

2. Table 1 could be more precise in giving the reference for the value of each parameter

3. L13: Five dengue serotypes ?

4. The conclusions refer sometimes broadly to "vector control" whereas the model only considers the SIT intervention (or genetic based methods similar to it) which is a rather new and specific intervention compared to vector control in general.

5. I would suggest to include some literature on the use of optimal control theory for epidemiological models in the main text (and not only the supplement). Some more details on the technicalities of the method which are quickly introduced in the conclusions (such as weighting, sensitivity) could also be added (maybe in the supplement) to better inform readers interested in this methodology.

6. In the code provided online: could you explain why two separate scripts are available?

Author's Response to Decision Letter for (RSOS-181843.R1)

See Appendix B.

RSOS-181843.R2 (Revision)

Review form: Reviewer 2

Is the manuscript scientifically sound in its present form?

Yes

Are the interpretations and conclusions justified by the results?

No

Is the language acceptable?

Yes

Do you have any ethical concerns with this paper?

No

Have you any concerns about statistical analyses in this paper?

No

Recommendation?

Accept with minor revision (please list in comments)

Comments to the Author(s)

I thank the authors for the improvement on the clarity of the manuscript. However, not all of my concerns regarding the representation of dengue were addressed. I try to reformulate them in more details below. These elements are important in case the paper is aimed at an audience larger than the one of mathematicians.

1. I understand that the important result is not the absolute costs. However, the choice of the objective function is a fundamental assumption in any optimization problem, that requires discussion.

- For example, as mentioned by the authors, 15 and 6, ref4 indicates that only 96 million out of 390 million infections are apparent. And if I understand correctly, not all the apparent infections are severe. My question is therefore, does the parameterisation of the model represent apparent infections or all infections?

In case it represents all infections, by considering that e.g. 25% of dengue cases are apparent, and only a fraction of them result in hospitalisation, then the overall cost of infections would decrease while the costs of interventions would remain the same. I was therefore interested to know whether the statement 1428-431 "A key finding, throughout all the scenarios we investigated, is how minimising the infected population is always of the greatest economic importance. As shown in Fig 3, even when increasing the cost of vector control to a far higher value than is realistic, any control method is always financially advantageous." was still valid under other assumptions for the hospitalisation proportion (e.g. much lower number of severe infections).

- As mentioned in the introduction, secondary infections may be more likely to lead to hospitalisation than primary infections (e.g. due to ADE). A different weighting of primary and secondary infections in the cost function might lead to different qualitative results. For this reason, it would be interesting to know the proportion of primary and secondary infections that occur in this outbreak (e.g. without interventions).

2. In my opinion, the choice to study a single short outbreak following the simultaneous introduction of 2 serotypes in a naïve population is a particular case of a dengue epidemic (in many dengue endemic areas, the disease pattern is characterised by recurring outbreaks and important serotype-specific herd immunity). This assumption should be highlighted more when formulating the conclusions.

In addition, I was wondering to which extend a 2-serotype model with cross-immunity is required to study a single epidemic outbreak in a naïve population. How many secondary infections occur? Accounting for the fact that the timescale of the epidemic (80 days) is smaller than the cross immunity period (120 days), I would suspect that the only secondary infections that occur are due to the assumption on exponential time in the infected compartment.

If not the dynamics among compartments, at least the proportion of individuals in the different compartments at the end of the period could be indicated (and the total number of infected over the studied period), to provide more context to the reader.

To this respect, other sentences would require clarification:

What does "the end of the model cascade" (1177) mean?

The sentence line 524-525-526 in the discussion seems to relate to this as well, but sounds unclear.

2. I don't understand how the population can remain constant if some individuals die because of the disease.

3. My remark concerning the 5th serotype was meant to call for more nuance in the phrasing of the sentence, because, even though some publications report the discovery of a 5th serotype, 4 serotypes are more commonly reported in the literature.

Decision letter (RSOS-181843.R2)

06-Mar-2020

Dear Mr Rawson:

On behalf of the Editors, I am pleased to inform you that your Manuscript RSOS-181843.R2 entitled "Optimal control approaches for combining medicines and mosquito control in tackling dengue" has been accepted for publication in Royal Society Open Science subject to minor revision in accordance with the referee suggestions. Please find the referees' comments at the end of this email.

The reviewers and Subject Editor have recommended publication, but also suggest some minor revisions to your manuscript. Therefore, I invite you to respond to the comments and revise your manuscript.

- Ethics statement

- Data accessibility

<http://datadryad.org/submit?journalID=RSOS&manu=RSOS-181843.R2>

- Competing interests

- Authors' contributions

AB carried out the molecular lab work, participated in data analysis, carried out sequence alignments, participated in the design of the study and drafted the manuscript; CD carried out the statistical analyses; EF collected field data; GH conceived of the study, designed the study,

coordinated the study and helped draft the manuscript. All authors gave final approval for publication.

- Acknowledgements

- Funding statement

Because the schedule for publication is very tight, it is a condition of publication that you submit the revised version of your manuscript before 15-Mar-2020. Please note that the revision deadline will expire at 00.00am on this date. If you do not think you will be able to meet this date please let me know immediately.

on behalf of Mark Chaplain (Subject Editor)
openscience@royalsociety.org

Reviewer comments to Author:
Reviewer: 2

Comments to the Author(s)

I thank the authors for the improvement on the clarity of the manuscript. However, not all of my concerns regarding the representation of dengue were addressed. I try to reformulate them in more details below. These elements are important in case the paper is aimed at an audience larger than the one of mathematicians.

1. I understand that the important result is not the absolute costs. However, the choice of the objective function is a fundamental assumption in any optimization problem, that requires discussion.

- For example, as mentioned by the authors, 15 and 6, ref4 indicates that only 96 million out of 390 million infections are apparent. And if I understand correctly, not all the apparent infections are severe. My question is therefore, does the parameterisation of the model represent apparent infections or all infections?

In case it represents all infections, by considering that e.g. 25% of dengue cases are apparent, and only a fraction of them result in hospitalisation, then the overall cost of infections would decrease while the costs of interventions would remain the same. I was therefore interested to know whether the statement 1428-431 "A key finding, throughout all the scenarios we investigated, is how minimising the infected population is always of the greatest economic importance. As shown in Fig 3, even when increasing the cost of vector control to a far higher value than is realistic, any control method is always financially advantageous." was still valid under other assumptions for the hospitalisation proportion (e.g. much lower number of severe infections).

- As mentioned in the introduction, secondary infections may be more likely to lead to hospitalisation than primary infections (e.g. due to ADE). A different weighting of primary and secondary infections in the cost function might lead to different qualitative results. For this reason, it would be interesting to know the proportion of primary and secondary infections that occur in this outbreak (e.g. without interventions).

2. In my opinion, the choice to study a single short outbreak following the simultaneous introduction of 2 serotypes in a naïve population is a particular case of a dengue epidemic (in many dengue endemic areas, the disease pattern is characterised by recurring outbreaks and important serotype-specific herd immunity). This assumption should be highlighted more when formulating the conclusions.

In addition, I was wondering to which extend a 2-serotype model with cross-immunity is required to study a single epidemic outbreak in a naïve population. How many secondary infections occur? Accounting for the fact that the timescale of the epidemic (80 days) is smaller

than the cross immunity period (120 days), I would suspect that the only secondary infections that occur are due to the assumption on exponential time in the infected compartment. If not the dynamics among compartments, at least the proportion of individuals in the different compartments at the end of the period could be indicated (and the total number of infected over the studied period), to provide more context to the reader.

To this respect, other sentences would require clarification:

What does “the end of the model cascade” (l177) mean?

The sentence line 524-525-526 in the discussion seems to relate to this as well, but sounds unclear.

2. I don't understand how the population can remain constant if some individuals die because of the disease.

3. My remark concerning the 5th serotype was meant to call for more nuance in the phrasing of the sentence, because, even though some publications report the discovery of a 5th serotype, 4 serotypes are more commonly reported in the literature.

Author's Response to Decision Letter for (RSOS-181843.R2)

See Appendix C.

Decision letter (RSOS-181843.R3)

23-Mar-2020

Dear Mr Rawson,

It is a pleasure to accept your manuscript entitled "Optimal control approaches for combining medicines and mosquito control in tackling dengue" in its current form for publication in Royal Society Open Science. The comments of the reviewer(s) who reviewed your manuscript are included at the foot of this letter.

on behalf of Prof Mark Chaplain (Subject Editor)
openscience@royalsociety.org

Appendix A

Dengue Paper Reviewer Comments

This manuscript takes the model from Hendron and Bonsall 2016 on "The interplay of vaccination and vector control on small dengue networks" and uses optimal control theory to address the same questions. This manuscript appears to come up with many of the same conclusions as the earlier paper.

The original Hendron and Bonsall (2016) paper develops and presents the model and uses it to present some initial hypotheses on the combined effects of multiple disease prevention methods, and the effect of multiple population centres. This study indeed strengthens these hypotheses with a more robust mathematical investigation, by utilising a full optimal control analysis, in comparison to the original paper's general sensitivity analysis.

My main issue with the paper was that neither the Hendron and Bonsall model, nor optimal control theory, were explained quite well enough in this manuscript to quite understand all of the details.

We have addressed the specific points raised below and have improved the mathematical details.

In particular, in the model (equations 1-9), it's difficult to understand what is being assumed in all of the terms involving the symbol capital Omega, which is meant to be the total net movement of hosts into and out of settlement i. More explanation is needed here (e.g. why is it $(N_i + \Omega_i)$ in the denominator?

We appreciate that the wording may not have been clear for explaining the Omega terms. Migration is not a permanent effect in the model, instead it is assumed that individuals may move to another settlement during the day, for example to commute to work in a city. The Omega terms encapsulate the change to the population during the day, those who have migrated out of a settlement, and those who have migrated in. As such, $(N_i + \Omega_i)$ represents the total population that is able to be bitten in settlement i during the day, including those who do not natively reside there. This allows for the case where someone from settlement j could migrate to settlement i, contract dengue, return to settlement j, and then further infect the vector population in settlement j. Hence we must divide our transmission rates by $(N_i + \Omega_i)$ to scale the infection rates by the particular class' magnitude with respect to the total population. E.g. Infection with dengue, moving from state S_i to I_i would happen at a rate proportional to $S_i / (N_i + \Omega_i)$. Logically, the smaller the population of S_i , the lower the chance of being bitten, and moving to class I_i .

We have added a paragraph to the manuscript at L115-123 to clarify this, and have amended some of the writing throughout.

*Similarly, although it may be explained in the Hendron and Bonsall paper, I really couldn't figure out what the $(Z_i / (Z_i + \psi_i * u_2))$ terms in equation 11 were assuming for the juvenile mosquitoes.*

We have added several lines at L151-158 to explain how this term operates. In short, it works very similarly to the above, in that the rate of juvenile births is reduced by a factor of $(Z_i / (Z_i + \psi_i * u_2))$, i.e. a percentage of how many of the existing adult mosquitoes there are against the total population including the released sterile mosquitoes. The more sterilised mosquitoes there are, the more unsuccessful reproductions, and the less juvenile births.

*Also in Equation 11, why is the form $J_1 * \ln(1 + \delta * J_1)$ used for the density-dependence term? That seems like a rather unusual form for intraspecific competition.*

This is a well-known function for describing density dependent processes acting on mortality. The derivation of this (and a range of other models for density dependence) have been well-researched (e.g. Bellows 1981). This form of density dependence is an effective descriptor of monotonic growth to a fixed point. We have included a relevant reference of Bellows (1981) at the point when the term is introduced (L160) to provide further discussion and

comparison between multiple forms of density dependence.

Line 154: Optimal control: Appendix S1 gives a very brief, and general, explanation for optimal control, but it seems that few more details should go into the main text, in order for the readers to be able to understand it. For example, in the paragraph starting on line 171, the term adjoint functions is not defined, and it is not explained that g , the governing ODEs, are simply the model defined above.

We aim to keep the mathematical details to a minimum in the main text of the paper and have explained the optimal control approach in the Appendix. It is certainly true that more information and definitions can be defined in the manuscript itself. As such we have added several lines throughout the optimal control section to give more context to the section and introduce the relevant terms.

Line 165: The explanation about using the square of the variables in the cost function is less than convincing. It simply says that it avoids incorrectly assuming a linear relationship. But, could it instead be incorrectly assuming a quadratic relationship (without a linear term)?

A linear control term in the objective functional greatly limits the amount of information that can be derived from our control variables. Using a linear control here would limit our controls to piecewise continuous “bang-bang” control. We had originally intended to refrain from including this detail, but agree that this instead creates more confusion, so we have added further explanation to the manuscript and a relevant reference for further information on this choice.

Line 184: It seems like a very strange assumption to have people travel to the central settlement, but never travel back again to their home settlement.

It is important to appreciate that this is not a permanent migration. All hosts have a “home node” which they always return to. Instead, some percentage of the node’s population may migrate temporarily to another settlement, creating a scenario whereby an individual could move to another node, contract dengue, return home, then infect the vector population at his home node. This line has been extended to explain this more clearly.

Figures 3-9: Need more explanation on how to interpret the figures.

For example, is Figure 3 just saying that it’s useful to perform vector control, releasing all possible SIT modified mosquitoes, up to some time point (50 to 80 days in the outbreak, depending on the cost), and then not so much? But, can we tell from this figure how helpful vector control actually is from the figure?

Further explanation has been added to the results section accompanying the figures.

In order to interpret the effect of the timing of the release, would it be useful to also show what’s happening to the disease outbreak, i.e. to show a trajectory of the population dynamics from the model to show what’s happening on day 50ish of the outbreak, with and without control?

The aim of this work is to establish an optimal control framework for investigating diseases (like dengue) on small networks. Capturing the multi-variable impact of the control changes would require additional investigations (beyond the scope of the current work) which we feel could draw attention away from the main findings of the paper. The basic changes to these variables can already be found in Hendron and Bonsall (2018).

It also might be useful to label the y-axis not just u_2 , but also include the definition of u_2 “Proportion of total possible SIT modified mosquitoes released”, so that it can be instantly interpreted. (I have the same comment for all of the subsequent figures).

In general, this has been amended for the majority of the results. However it is inappropriate for all figures given the way in which the results are presented.

line 207 states “Evidently, the release of modified mosquitoes is a powerful technique”. How can we tell this from the figure? The figure just shows that the optimal strategy is to use the modified mosquitoes, but it doesn’t show how much it actually reduces the disease outbreak.

As mentioned above, to show the changes made across all variables would require a considerable number of additional figures. We have however altered the wording to better align with the presented figure, instead referring to the release of modified mosquitoes as a 'highly desirable' technique.

Line 313: I'm confused by the statement "as migration is the sole source of infection for these outer towns". How does this work? I thought that movement was only from the towns to the central city.
The towns in this case study are initialised free from infection. As such they can only become infected by migrating to the central hub, becoming infected, and then returning to the outer town to infect the vector population, facilitating the spread of disease. The line has been amended to better explain this effect, and the earlier clarifications to the effect of migration should further help explain this phenomena.

Line 440: This paragraph is hard to understand without more details of how the optimal control optimization works.

The line exists primarily as an interesting sidenote to the reader who is more familiar with optimal control methods. Understanding the choice of bias parameters here is not required to understand the outcomes of the study, however it would likely be of some interest to certain readers.

Throughout the manuscript: A canonical set of parameters is used throughout. Some of these parameters have a surprising level of precision (e.g. 0.0541). How much do the conclusions of the manuscript depend on the exact choice of these parameters?

Parameter variation was not considered within the scope of this paper. As noted the aim of this work was to establish an optimal control framework approach for dengue on small networks. Additional work is underway to investigate the full variation in parameters within and between nodes. To include this in the current work would be unduly complex and we are minded to develop this into a related but independent piece of research.

Minor issues:

Abstract: There's an extra "that" in the sentence: "We show that that a combination..."

This has been amended.

Line 430: extra ",," in: "before rejoining ,one"

This has been amended.

Appendix B

Reviewer comments to Author:

Reviewer: 2

Comments to the Author(s)

In this paper, Rawson et al. use optimal control theory on a previously published dengue model, in order to identify the best combination of vaccination and sterile insect release to reduce the number of infections.

I recognise the interest and usefulness of the optimal control methodology for epidemiological models; however, I have a series of **major concerns regarding the hypotheses** on dengue dynamics in both the model and the objective function, which may affect the conclusions, especially those formulated in terms of policy recommendations.

We are very grateful for the comprehensive feedback kindly provided. On reflection of these comments we acknowledge that greater clarity can be provided as to the precise framing of the model, and the specific insight that the optimal control results can provide. We have addressed the points below with further clarification or amendments to the manuscript. All tracked changes are displayed in red in the newly submitted manuscript. Our objectives and findings are now better made clear, and we appreciate the insight into where further explanation or justification was required.

Major comments.

1. The model aims to minimize the total number of infections and associates a hospitalization cost to each infection. In dengue epidemics, there is potentially a very large proportion of asymptomatic or mildly symptomatic cases, that would not require hospitalization. Unless the current model parameterization represent the dynamics of severe cases only, this hypothesis in the cost function may not be realistic for dengue.

It is certainly true that the current choice of our constant C_1 in the cost function currently only captures the economic cost of infection due to hospitalisation. However, our model intends to capture not just the economic importance of preventing dengue infection, but the innate desire to prevent infections occurring. It is always a difficult balance when deciding on which weightings to apply to a cost function, to capture both the objective and subjective desires that the model attempts to convey. As such we consider each infected individual as having the **potential** to incur the cost of C_1 , hence the included formulation. The reviewer is correct in that this has the potential to over-predict the exact economic burden of infection, but such a value is extremely difficult to predict with accuracy. Indeed, the study by Carrasco et al. (2011) referenced in the manuscript concluded an annual cost ranging between \$0.85 billion and \$1.15 billion; a \$300million window of uncertainty, not to mention the vast differences in costs ranging from country to country. Indeed, while we could be over-estimating the economic cost of each dengue case, we could also risk under-estimating by not factoring for the economic cost of lost labour hours. The important result of this work is not the exact final cost values we arrive at, but the dynamics of optimal treatment. These dynamics themselves are fundamentally unchanged by the exact choice of cost values, as depicted in Figures 3 – 7, where we have varied cost parameters to display that the fundamental dynamics are unchanged. We can certainly agree however that the decision behind this choice was poorly conveyed in our initial submission, and we have extended our justification of our cost value for C_1 in light of this.

2. **Surprisingly, disease induced mortality is included in this model without demography** (does the human population stay constant?), but not in the cost function, giving the impression that deaths are not accounted for in the optimization of intervention strategies.

Firstly, it is true that we do not consider or perform case studies to account for demographic differences between serotypes. It was felt that with four distinct case studies already, the inclusion of a case study investigating differences between serotype capability was unnecessary, as our primary result is focussed around the relationship between combinations of disease treatment, while already considering the impact of multiple settlements and antibody-dependent enhancement. Alterations to the serotypes' individual responses to vector control also does not affect the overall dynamics of the optimal response, only the intensity of the control application.

Secondly, our human population does stay constant for this model, which we discuss further in our response to comment 3.

Disease induced mortality events are not themselves specified in the cost function, but are captured by the inclusion of the infected population. Indeed, the number of disease-induced deaths will always be a proportion of the infected population (α_1 and α_2), and so optimising upon both deaths and infections is no different to optimising upon just infected individuals, differing only by choices in cost value. There is a linear relationship between the infected class and the number of disease-related deaths. In treating any epidemic, policy seeks to minimise the number of infected individuals, as this will of course minimise the number of deaths by proxy. As mentioned in our response to comment 1, any added cost incurred by a death event is encapsulated by the already wide window of uncertainty surrounding any cost value we employ. To better clarify this effect, we have added further explanation at the initial presentation of the cost function.

3. The time indices over which the objective function is integrated (0 and t_f) are not clearly defined: because they represent the period of time over which costs are calculated, they may impact the results of the optimization and should therefore be detailed and discussed. In addition, I do not understand what the time dimension in the figures refers to (introduction of interventions? Duration of intervention? Time period considered in the objective function?).

Much of the confusion here was born from our mistake in not properly clarifying the timescale within the manuscript. We included in our axis labels in our figures that time is modelled in days since model initialisation, but did not specifically state this in the manuscript, which has now been amended. The reviewer is correct in that, for growing and evolving populations, the costs will vary depending on the length of the experiment, due to the oscillating number of infected individuals. However, as noted in comment 2, we consider only a fixed population within our case studies, as the timescale of the model is considered only in days, making human birth events a negligible inclusion. As such, our case studies capture cases where the population has completed the cascade to either death or recovery. As a result, the control interventions then drop to 0 (as seen in Figures 3-9) meaning the total cost will remain fixed beyond this point. This means that integrating from 0 to $t_f = 80$

(as done in the manuscript) would be no different than integrating from 0 to $t_f = 1000$. A discussion and explanation of this has been added to the methods section.

4. The hypotheses on vaccine effect do not seem to represent the current Dengvaxia vaccine, which, to my knowledge, is only recommended for individuals with pre-existing dengue immunity, and not for naïve individuals (S).

An effective dengue vaccine that provides balanced protection is still elusive, and the controversy surrounding Dengvaxia is evolving rapidly, with many ensuing complications having occurred between now and initial submission (since August of last year, the Philippines Department of Health has banned the use of Dengvaxia).

Due to ongoing study into the long-term efficacy and viability of Dengvaxia we specifically wanted to avoid representing one specific medicine. As such, we instead explore the impact of varying drug intervention, with Case 2 specifically investigating the impact of uncertain vaccine application and efficacy. Since we cannot represent the current and developing use of the Dengvaxia vaccine, we instead consider application to the naïve population, representing the overall objective of vaccine treatment, providing resistance before infection occurs. The variability and uncertainty in vaccine capability is directly captured within our results in Case 2.

We feel it is important to not be interpreted as a precise application of Dengvaxia, and have included the above discussed points within the manuscript to avoid misinterpretation.

5. The initialisation of the model in a naïve population ($R=0$, $S_a=0$, $S_b=0$, $R_a=0$, $R_b=0$) does not reflect current recommendation for the vaccine, which would not be administered in a naïve population, but in a population with high endemicity. In addition, the text mentions that the dynamic of the model consists in one "disease outbreak" (L256): does this mean the studied trajectory does not include oscillations or endemicity? A graphic representing the disease dynamics (e.g. number infected individuals over time) should be included to clarify this.

As mentioned in the above comments, our consideration of a fixed host population means that the model considers the timespan of a fixed disease event, hence the use of the term of "disease outbreak". Also, concerns surrounding the representation of vaccine coverage are discussed in our response to comment 4, however consideration of endemic population infection is explored in Case 3 of our study.

We are hesitant to include a figure depicting the infected population over time, as the information conveyed by this is minimal. It would merely depict the decreasing infected population over time in the endemic case, and represent minimal individuals in other cases

as vaccines rapidly move the population to the recovered class. This then reaches zero as the population finally fully inhabits the recovered class. We have included further clarification as to the use of a fixed human population and feel that this should suffice in explaining the human population dynamics, however we can include such a figure if the reviewer still feels this would be helpful in light of the further clarifications.

6. I understand that the main objective of the manuscript is not to perform a sensitivity analysis; however, the model relies on many parameters and assumptions that are neither varied nor even discussed. This is even more important for some parameters which, to my knowledge, are not precisely quantified and still debated in the literature (duration of cross-immunity, some mosquito parameters, people's movements)

The initial publication of the model includes a sensitivity analysis across the model parameters, revealing no impact to overall model dynamics due to parameter variation. As such, it was not necessary to further vary upon parameter value as well as cost values. This is certainly an important justification to provide, which we have now included.

Minor comments

1. The writing of the equations could be simplified to ease the reading (merge the sums, simplify the notations for parameters which take the same value throughout the paper (e.g. ρ_1 and ρ_2), change the notation g which is used twice)

Merging of the sums would create a longer expression for all of the equations of the model, as well as arguably reducing the ease of comprehension, and no longer being immediately comparable to the original model publication.

We feel compiling ρ_1 and ρ_2 makes distinguishing elements of the Hamiltonian and the adjoint equations more difficult, as well as frustrating cross-reference with the original model presentation.

When referring to the governing ODEs in our Hamiltonian definition, we have changed our notation to prevent confusion.

2. Table 1 could be more precise in giving the reference for the value of each parameter. References have been labelled to the respective values.

3. L13: Five dengue serotypes ?

Yes. While four serotypes are more commonly observed, a fifth serotype was reported in 2015: Mustafa et al. (2015) - "Discovery of fifth serotype of dengue virus (DENV-5): A new public health dilemma in dengue control".

4. The conclusions refer sometimes broadly to "vector control" whereas the model only considers the SIT intervention (or genetic based methods similar to it) which is a rather new and specific intervention compared to vector control in general.

We had used the terminology of 'vector control' within the more focused specification of SIT, but upon re-reading our Discussion, it is clear that we overuse this term to the point of

causing confusion. As such we have added a line to the start of our discussion reiterating that we are considering the release of modified mosquitoes as a form of vector control, not to be confused with the use of insecticides. We have also reduced the reference to “vector control” by replacing instead with the term “modified mosquito release”. Note however that the dynamic effect on host infections caused by reduced mosquito populations is still relevant in the context of wider vector control methods.

5. I would suggest to include some literature on the use of optimal control theory for epidemiological models in the main text (and not only the supplement). Some more details on the technicalities of the method which are quickly introduced in the conclusions (such as weighting, sensitivity) could also be added (maybe in the supplement) to better inform readers interested in this methodology.

We have moved the motivating literature references and discussion to the main text. Extra information regarding weighting is very slight, and so has just been added to the main text.

6. In the code provided online: could you explain why two separate scripts are available? We have updated the readme file to explain better. Both scripts perform the same task, one is slower but easier to follow, while the other is optimised for runtime, but harder to interpret at first read.

Appendix C

Comments to the Author(s)

I thank the authors for the improvement on the clarity of the manuscript. However, not all of my concerns regarding the representation of dengue were addressed. I try to reformulate them in more details below. These elements are important in case the paper is aimed at an audience larger than the one of mathematicians. We thank the reviewer for taking the time to highlight the areas requiring further information, clarity and motivation. We appreciate the effort made in making the manuscript as accessible as possible, and we have addressed the remaining concerns.

1. I understand that the important result is not the absolute costs. However, the choice of the objective function is a fundamental assumption in any optimization problem, that requires discussion.

- For example, as mentioned by the authors, 15 and 6, ref4 indicates that only 96 million out of 390 million infections are apparent. And if I understand correctly, not all the apparent infections are severe. My question is therefore, does the parameterisation of the model represent apparent infections or all infections?

We thank the reviewer for this point. To clarify, our model represents all infections, as even asymptomatic host infections have an important role to play in the spread of the disease by further infecting more vectors and, by proxy, more hosts.

In case it represents all infections, by considering that e.g. 25% of dengue cases are apparent, and only a fraction of them result in hospitalisation, then the overall cost of infections would decrease

while the costs of interventions would remain the same. I was therefore interested to know whether the statement I428-431 "A key finding, throughout all the scenarios we investigated, is how minimising the infected population is always of the greatest economic importance. As shown in Fig 3, even when increasing the cost of vector control to a far higher value than is realistic, any control method is always financially advantageous." was still valid under other assumptions for the hospitalisation proportion (e.g. much lower number of severe infections).

On this point, there is indeed a wide degree of uncertainty in the exact cost value to attribute to the infected class. If indeed we say 25% of all infections resulted in hospitalisation, it could be argued to multiply our cost value C_1 by 0.25. This is theoretically equivalent to weighting the "importance" of minimising the infected class.

Simultaneously however, one could argue that an infected individual also represents a further economic cost by way of their ability to then infect others. Likewise one could argue for a weighting to increase the C_1 value to represent the social cost of infection, and the innate desire to actively prevent infections. The question then of whether our results are still valid for varying levels of hospitalised proportions is equivalent to asking if our results are still valid for varying values of C_1 . Because the variables of our objective functional are not multiplied together at any point, the same dynamic behaviour would be observed from increasing C_1 as it would by decreasing C_2 . We observe a constant "tug-of-war" effect between trying to minimise the infected class without over-spending in control measures. As such, the reviewer is absolutely correct that variations to these values could hide important implications for the results, which is why all of our

figures included results for multiple cost values, to explicitly show that the dynamical behaviour is preserved. The decision was made to vary the cost value for the control measures (C_2 and C_3), as opposed to C_1 as these variables are considered to be those which have most degrees of freedom. More importantly we did not wish to potentially conflate the results by seeming to “alter” parameters related to what we previously introduced as unalterable “state variables”. Evidently we may have ended up creating further confusion. To address this we have added a paragraph to our ‘Optimal Control Problem’ section to clarify this properly, and to also make clear why it is good practice to vary on cost values to ensure results are consistent. (see L234 – L249).

- As mentioned in the introduction, secondary infections may be more likely to lead to hospitalisation than primary infections (e.g. due to ADE). A different weighting of primary and secondary infections in the cost function might lead to different qualitative results. For this reason, it would be interesting to know the proportion of primary and secondary infections that occur in this outbreak (e.g. without interventions).

It was considered at one point to include separate, higher, cost values for secondary infections. However, with an already high degree of uncertainty in hospitalisation costs, it was unclear from the literature how best to scale this “greater” cost. When initial tests revealed that scalings were equivalent across cost values (meaning no difference to qualitative results), and given the already overwhelming size of the ensuing adjoint equations, the decision was made to simplify the objective functional to include all infections under the same cost value. We were careful to avoid too much discussion as to the

underlying model dynamics, so as to distinguish the work from the original model publication, but have included in our added paragraph clarification that secondary infections display similar dynamics, but always at a lower amount compared to primary infections. We have also added a figure of model outputs to help address these points in appendix S4.

2. In my opinion, the choice to study a single short outbreak following the simultaneous introduction of 2 serotypes in a naïve population is a particular case of a dengue epidemic (in many dengue endemic areas, the disease pattern is characterised by recurring outbreaks and important serotype-specific herd immunity). This assumption should be highlighted more when formulating the conclusions.

We have added this insight to our 'Coda' section to help highlight this.

In addition, I was wondering to which extend a 2-serotype model with cross-immunity is required to study a single epidemic outbreak in a naïve population. How many secondary infections occur? Accounting for the fact that the timescale of the epidemic (80 days) is smaller than the cross immunity period (120 days), I would suspect that the only secondary infections that occur are due to the assumption on exponential time in the infected compartment. If not the dynamics among compartments, at least the proportion of individuals in the different compartments at the end of the period

could be indicated (and the total number of infected over the studied period), to provide more context to the reader.

On this helpful suggestion we have included a new supplementary figure of the state variable outputs for the case considered in Figure 4, when all control aspects are first considered. Here it can be seen that the secondary infections are considerably lower than the primary infections, so as to be practically negligible under the optimal control response.

To this respect, other sentences would require clarification:

What does "the end of the model cascade"(l177) mean?

We have changed this wording to "...not be affected by the simulated time period, as we consider the total time until all hosts have reached a model state whereby they can no longer be aided by control measures."

The sentence line 524-525-526 in the discussion seems to relate to this as well, but sounds unclear.

We have modified this line, as the following lines expand on our point.

2. I don't understand how the population can remain constant if some individuals die because of the disease.

Our apologies for the confusion, we mean to say that the initial population does not increase, in that birth events are not considered given the scale of our case studies.

3. My remark concerning the 5th serotype was meant to call for more

nuance in the phrasing of the sentence, because, even though some publications report the discovery of a 5th serotype, 4 serotypes are more commonly reported in the literature.

In light of this we have clarified in L14-L16 that much of the literature may only refer to 4 serotypes.